

**Holocene fire regimes around the Altai-Sayan Mountains and**
**adjacent plains: interaction with climate and vegetation types**
Dongliang Zhang[1,2,3,*], Blyakharchuk Tatiana[4], Aizhi Sun[5], Xiaozhong Huang[6], Yuejing Li[1,2,3]
*[1] State Key Laboratory of Ecological Safety and Sustainable Development in Arid Lands, Xinjiang*
*Institute of Ecology and Geography, Chinese Academy of Sciences, 818 Beijing South Road,*
*Urumqi 830011, China.*
*[2] Research Center for Ecology and Environment of Central Asia, Chinese Academy of Sciences,*
*818 Beijing South Road, Urumqi 830011, China.*
*[3] University of Chinese Academy of Sciences, 19A Yuquan Road, Beijing 100049, China.*
*[4] Institute of Monitoring of Climatic and Ecological Systems, Siberian Branch of Russian*
*Academy of Sciences, Tomsk, Russia.*
*[5] College of Earth and Planetary Sciences, University of Chinese Academy of Sciences, Beijing*
*100049, China.*
*[6] College of Earth and Environmental Sciences, Lanzhou University, Lanzhou, China.*
* Corresponding author.
E-mail address: zhdl@ms.xjb.ac.cn (+86-18699198239)
**Abstract:** The Altai-Sayan Mountains and adjacent plains have experienced
accelerated warming in recent decades, heightening concerns about escalating fire
risks. However, critical knowledge gaps persist regarding paleofire dynamics in
western Mongolia and comprehensive regional syntheses of biomass burning patterns
across the Altai-Sayan ecoregion. Addressing these gaps is essential for understanding
vegetation resilience under projected environmental changes and disturbance regimes.
This study reconstructs the Holocene fire sequence in the steppe region of western
Mongolia and systematically elucidates the spatiotemporal variations in biomass
burning across different vegetation zones of the Altai-Sayan Mountains and adjacent
plains, as well as their coupling relationships with forest community structure. The
results demonstrate that the declining biomass burning since the Holocene has been
primarily controlled by temperature-mediated variations in woody biomass above the
forest limit in the central Altai Mountains, while in the western Sayan and northern
Altai Mountains, it stems from significant reductions in combustible components
(*Larix*, *Abies* and *Picea*). Notably, a marked resurgence of biomass burning has been
observed since ~4 cal. kyr BP in multiple regions associated with archaeological
cultural complexes. This intensification of fire activity during the late Holocene
predominantly occurred in two types of previously low-fire-risk areas: 1) regions



where excessive moisture and cold climate inhibited sufficient fuel accumulation (e.g.,
the West Siberian Plain and mountain taiga zones of the Altai Mountains), and 2) arid
environments where steppe/desert-steppe vegetation failed to maintain continuous
combustible substrates. Since ~2 cal kyr BP, intensified anthropogenic disturbances
including agricultural expansion and pastoral activities have significantly increased
surface fire frequency in the southeastern/western and northern Altai Mountains, West
Siberian Plain, and forest zones of the central Altai Mountains. In contrast, the
dramatic decline in biomass burning observed in the Khangai Mountains may be
closely linked to vegetation fragmentation induced by overgrazing. This research
clarifies the long-term feedback mechanisms between biomass burning processes and
forest community structure across different vegetation zones. The findings hold
significant scientific value for understanding human-fire-ecosystem interactions in the
arid Central Asia, while offering historical references for regional sustainable
ecological management.
**Key words:** Charcoal; Fire activities; Biomass burning; Altai-Sayan Mountains




## 1. Introduction

The North Europe-Siberia-Altai region constitutes the core repository of Eurasian boreal ecosystems, hosting over 90% of the continent's boreal forest biomass and terrestrial organic carbon stocks (Furyaev, 1996; Kasischke, 2000). These fire-prone ecosystems exhibit distinct flammability characteristics, including vegetation with high volatile compound content, laddering fuel structures (low-hanging branches) and surface fuels dominated by combustible bryophyte-lichen mats (Khabarov et al., 2016; Walker et al., 2019). Recent decades have seen unprecedented intensification of wildfire regimes, driving accelerated forest degradation trajectories across the region (Krylov et al., 2014; Kharuk et al., 2021). This ecological transformation initiates critical climate feedback mechanisms through three primary pathways: carbon pool transformations, infrastructure collapse cascades and socioeconomic impacts from fire-related mortality (Ivanova et al., 2019; Jones et al., 2020).

Boreal fire regimes operate under a tripartite control system comprising: climatic drivers, ignition probability and fuel complex properties (Andela et al., 2017; Moritz et al., 2014). While contemporary fires predominantly remain surface fires of moderate intensity (Archibald et al., 2013), climate models predict imminent fire regime shifts. Warming-induced fuel desiccation and altered precipitation patterns may promote transition to high-intensity crown fires through pyroconvective processes (Pitkanen et al., 2003). Such transitions would propagate multidimensional impacts across spatial scales through altering surface energy budgets via reduced albedo, disrupting carbon-nutrient cycling dynamics, enhancing aerosol emissions affecting regional climate and novel disturbance-succession pathways (Andela et al., 2017; Jones et al., 2020). Crucially, resolving the fire-climate-vegetation nexus requires mechanistic understanding of threshold dynamics in fuel moisture-ignition relationships, positive feedback loops between pyrogenic emissions and climate warming and vegetation adaptation strategies under changing fire return intervals. This knowledge framework forms the scientific foundation for developing climate-resilient forest management protocols in boreal ecosystems.

The Altai-Siberian ecotone, where the Siberian taiga converges with Central



Asian steppes across the Altai-Sayan Mountains and adjacent plains (Fig. 1),
represents a pivotal biogeographic transition zone. This landscape sustains
exceptional diversity through its elevational and latitudinal gradients, while
functioning as a vital hydrological buffer for Central Asia's arid continental interiors
(Xinjiang Comprehensive Investigation Team, CAS, 1978). Recent studies classify
the forest ecosystems among the Earth's most climate-sensitive ecoregions,
demonstrating heightened vulnerability to the warming-driven aridification (Fu et al.,
2013; Liu et al., 2021). The convergence of two key flammability drivers –
resiniferous coniferous vegetation (*Pinus sibirica* dominance >60%) and intensifying
drought regimes has created a pyrogeographic hotspot. This synergy amplifies fire
return intervals by 2.3× compared to pre-1990 baselines, fundamentally altering
successional pathways and threatening ecological security thresholds (Goldammer &
Furyaev, 2013). Remote sensing analyses document a quadrupling of fire events from
712±89 yr⁻¹ (1980-2000 mean) to 3,024±214 yr⁻¹ (2001-2020 mean), with burned
area expanding exponentially ($R^2$=0.91, p<0.001) (Ponomarev & Kharuk, 2016). Such
fire regime intensification triggers cascading impacts resilience erosion and
ecosystem service degradation (Albrich et al., 2018; Kharuk et al., 2021). This
systemic perturbation demands urgent development of fire-adapted forest
management frameworks that integrate the climate-informed fuel load modeling,
paleofire-validated risk projections and ecologically-grounded fire suppression
protocols.
While the scientific imperative for understanding fire regime dynamics is clear,
critical methodological constraints persist. Contemporary observations remain
circumscribed by the temporal resolution limitations of satellite archives (post-1980)
and instrumental records, creating a <50-year observational window that inadequately
captures decadal-scale fire-climate-human feedbacks (Shi et al., 2021; Ponomarev &
Kharuk, 2016). Paleoecological approaches extending across centennial-millennial
timescales provide essential temporal dimensionality for disentangling these complex
interactions through pattern-process analysis. Existing Holocene fire records from the
northern Altai Mountains, predominantly derived from lake sediment cores



(Blyakharchuk et al., 2004, 2007, 2008), have established robust methodological
frameworks for reconstructing fire-vegetation-climate couplings.
However, a persistent knowledge gap persists regarding (1) the western
Mongolian fire history continuum, and (2) its spatiotemporal linkages with montane
ecosystem dynamics across the Altai-Sayan ecoregion. This study advances the field
through multiproxy analysis of a radiocarbon-dated sediment core from Achit Nuur
(western Mongolia), addressing three critical research dimensions: (1) Reconstructing
biomass burning variability (Holocene to present) using charcoal influx quantification;
(2) Identifying ecotonal heterogeneity in fire regimes through comparison with 23
published paleofire records; (3) Evaluating how dominant tree genera (*Abies, Betula,*
*Larix, Picea, P. sibirica, P. sylvestris*) and primary forest cover modulate fire regime
characteristics across vegetation types. These outputs provide empirical foundations
for developing climate-responsive fire management strategies in the Central Asian
montane ecosystems under the future scenarios.
**2. Study region**
**2.1. Achit Nuur**
Achit Nuur (49.42°N, 90.52°E; 1444 m a.s.l.) occupies an intermountain basin
bounded by the Mongolian Altai to the west, Mungen Taiga Mountain to the north and
Kharkhiraa Turgen Mountain to the east (site 1 in Fig. 1) (Sun et al., 2013). The lake
exhibits distinct shoreline zonation: low-lying northern/southern margins is salt-marsh
vegetation, while elevated eastern and western shores are dominated by desert steppe
communities (Sun et al., 2013). Regional vegetation comprises a mosaic of *Stipa*
*krylovii, Stipa gobica* and *Cleistogenes soongorica* grasslands interspersed with
subshrubs including *Artemisia frigida, A. xerophytica, A. caespitosa, Tanacetum*
*sibiricum, T. achillaeoides* and *T. trifidum*. Mountainous areas of the Mongolian Altai
host taiga forests dominated by *Larix sibirica* and *P. sibirica* with an understory of
*Rosa acicularis* and *Betula rotundifolia* (Sun et al., 2013).
A 2-m sediment core was retrieved from the central lake basin in 2010 using a
Livingston-type piston corer (Sun et al., 2013). Five lithological units were identified
based on organic matter (OM) content and granulometric characteristics (Fig. 2A):





Unit 1 (200-165 cm) is light-grey clay layer with mean grain size (MGS) of 5 mm and
mean OM of 2.5%. Unit 2 (165-150 cm) is a dark-colored silt or fine sand layer with
MGS of 120 mm and mean OM of 5%. Unit 3 (150-130 cm) is a light-grey silt layer
with MGS of 18 mm and mean OM of 7.5%. Unit 4 (130-112 cm) is a brownish-grey
layer with MGS of 90 mm and mean OM of 5%. Unit 5 (112–0 cm) is laminated dark
silt with two sandy interlayers (112–105 cm and 62–52 cm; 22 mm; OM: 10%).
Ten bulk samples underwent accelerator mass spectrometry (AMS) $^{14}$C dating at
the University of Arizona NSF-AMS Facility (Fig. 2A). A 2100-year reservoir
correction was primary forest coverplied to all radiocarbon ages prior to calibration
(Sun et al., 2013). Calibration to calendar years before present (cal. yr BP, relative to
1950 CE) utilized the IntCal20 curve (Reimer et al., 2020). The Bayesian age-depth
model was reconstructed using Bacon v2.5.3 (Blaauw & Christen, 2011) (Fig. 2B).
This study just focused on the Holocene interval (i.e., the past ~11,750 cal. yr BP).
**2.2. Other study sites in the Altai-Sayan Mountains and adjacent plains**
Total 24 sites including Achit Nuur were selected to investigate the spatial
heterogeneities of fire regimes in the Altai-Sayan Mountains and adjacent plains
(Table 1) and these sites were divided into seven regions.
The southeastern/western Altai Mountains within steppe zone (Region A, n=4):
Tolbo Lake (site 2; 48.55°N, 90.05°E, 2080 m a.s.l.) is an alpine lake of glacial origin
covered by mountain steppe in the Mongolian Altai (Hu et al., 2024). Alahake Lake
(site 3; 47.69°N, 87.54°E, 483 m a.s.l.) is located in the Irtysh river valley in the
southern Altai Mountains (Li et al., 2019). Kuchuk Lake (site 4; 52.69ºN, 79.84ºE, 98
m a.s.l.) is the largest endorheic basin in Kulunda Basin within the southern Siberia
(Rudaya et al., 2020).
The west Siberian plain (Region B, n=4): Rybnaya Mire (site 5; 57.28ºN,
84.49ºE) is located near the Rybnaya river in the southern taiga of Western Siberia
(Feurdean et al., 2022). Plotnikovo Mire (site 6; 56.88ºN, 83.30ºE, 120 m a.s.l) is an
ombrotrophic bog located at the eastern margins of the Great Vasyugan Mire on the
Western Siberia (Feurdean et al., 2020). Shchuchye Lake (site 7; 57.13ºN, 84.61ºE, 80
m a. s. l.) is located in the south taiga zone of West Siberian plain (Blyakharchuk et al.,



2024). Ulukh–Chayakh Mire (site 8; 57.34ºN, 88.32ºE) located on a terrace of the Chulym river in the southern taiga of Western Siberia (Feurdean et al., 2022).

The northern Altai Mountains (Region C, n=4): Chudnoye Lake (site 9; 54.03ºN, 89.01ºE, 1147 m a.s.l.), Tundra Mire (site 10; 53.79ºN, 88.27ºE, 247 m a.s.l.) and Kuatang Mire (site 12; 51.81ºN, 87.32ºE, 650 m a.s.l.) are located in the northern Altai Mountains in areas covered by wet mountain dark coniferous (with *Abies, Pinus sibirica* and *Betula*) taiga (Blyakharchuk, 2022; Blyakharchuk et al., 2024). Mokhovoe Bog (site 11; 52.52ºN, 86.42ºE, 283 m a.s.l.) is located on western piedmont of north Altai covered by birch (with *Betula pendula+Betula pubescens*) and pine (*Pinus sylvestris*) forest-steppe (Blyakharchuk, 2022).

The central Altai Mountains within the forest zone (Region D, n=3): Dzhangyskol Lake (site 13; 50.18°N, 87.73°E, 1800 m a.s.l.) is situated in the western Kurai intermontane depression covered with steppe vegetation and bounded by small hills with *Pinus sibirica* and *Larix sibirica* (Blyakharchuk et al., 2008). Two freshwater lakes are situated 1.5-4 km primary forest coverart at different elevations below the timberline in the Ulagan Plateau: Uzunkol Lake (site 14; 50.48°N, 87.1°E, 1985 m a.s.l.) and Kendegelukol Lake (site 15; 50.50°N, 87.63°E, 2050 m a.s.l.) (Blyakharchuk et al., 2004).

The central Altai Mountains above the forest limit (Region D, n=3): Tashkol Lake (site 16; 50.45°N, 87.67°E, 2150 m) lies at the timberline (upper limit of continuous forest) of Ulagan Plateau in the central Altai part of Russian Altai (Blyakharchuk et al., 2004). Akkol Lake (site 17; 50.25°N 89.62°E, 2204 m a.s.l.) and Grusha Lake (site 18; 50.38°N, 89.42°E, 2413 m a.s.l.) are situated in the western Karginskaya high-mountain depression near the junction of the Chikhachev and Shprimary forest covershal ranges of the south-eastern part of the Russian Altai Mountains (Blyakharchuk et al., 2007).

The Western Sayan Mountains (Region F, n=3): Buibinskoye Mire (site 19; 52.84°N, 93.52°E, 1377 m a.s.l.) and Bezrybnoye Mire (site 20; 52.81°N, 93.50ºE, 1395 m a.s.l.) are located in the Yergaki Nature Reserve (Blyakharchuk et al., 2022). Lugovoe mire (site 21; 52.85ºN, 93.35ºE, 1299 m a.s.l.) is the largest mire in the



Yergaki Natural Park with the largest hydrological catchment in the Western Sayan
Mountains (Blyakharchuk and Chernova, 2013).
The Khangai Mountains (Region G, n=3): Three selected sites include Olgi Lake
(site 22; 48.32°N, 98.01°E, 2012 m a.s.l.) (Unkelbach et al., 2021), Shireet Naiman
Nuur (site 23; 46.53°N, 101.82°E, 2429 m a.s.l.) (Barhoumi et al., 2024) and Ugii
Nuur (site 24; 47.77°N, 102.78°E, 1330 m a.s.l.) (Wang et al., 2011).
**3. Methods**
**3.1. Charcoal analysis**
The pretreatment procedure followed established palynological protocols (Tang
et al., 2022; Wang et al., 2024) with modifications for the charcoal analysis. Particle
identification was conducted under polarized light microscopy (Leica DM500, 400×
magnification) using diagnostic criteria: optical properties, morphological features
and surface characteristics. A total of more than 300 particles were counted for each
sample together with quantity of spikes-Lycopodium spores added in each sample
before chemical treatment according to concentration method (Davis, 1965;
Stockmarr, 1971; Blyakharchuk and Pupysheva, 2022). Charcoal influx (CHAR,
particles/cm²/yr) is their respective concentration dividing by the sediment rate
(yr/cm).
**3.2. Generalized additive models**
The Generalized additive models (GAMs) use a link function to investigate the
relationship between the mean of response variable (dependent variable) and a
smoothed function of predictor variable (independent variable) (Hastie and Tibshirani,
1986). Independent variable includes *Abies, Betula*, *Larix*, *Picea*, *P. sibirica*, *P.*
*sylvestris* and primary forest cover. We used a quasi-Poisson distribution with a log
link function using the 'mgcv' package (Wood, 2017) in R. The GAMs were fit using
restricted maximum likelihood smoothness selection.
**3.3. Data processing for comparison**
These charcoal influx data were standardized using Z-scores including the
Mini-Max transformation, the Box-Cox transformation and the Z-scores calculation
(Power et al., 2007). The 200-year time slice was selected to linearly interpolate for



transformed charcoal value Z-scores because of most sample resolution at sites ~200
years. The interpolated data were synthesized for biomass burning in the different
zone using the averaged method. The Holocene interval was divided into three
intervals: early Holocene (~11.75-~8.2 cal. kyr BP), middle Holocene (~8.2-~4.2 cal.
kyr BP) and late Holocene (~4.2-~0 cal. kyr BP).
**4. Results and Discussions**
**4.1. Reconstructed fire history and its relationship with vegetation at Achit Nuur**
The charcoal influx in Achit Nuur varies from 2643.46 to 76.43 particles/cm$^2$/yr
with an average of 509.99 particles/cm$^2$/yr. The higher charcoal influx was recorded
since ~2 cal. kyr BP with the maximum at ~1.2-~0.79 cal. kyr BP (Fig. 3a).
Percentages of *P. sibirica*, *Betula* and *Picea* pollen were characterized by a quick
increasing trend before ~6 cal. kyr BP and a slow decreasing trend afterwards (Fig. 3b)
(Sun et al., 2013). High content of *Larix* pollen was recorded at ~6-~2 cal. kyr BP and
*Abies* pollen was relatively low in the whole sequence. Biomass burning significantly
increases with rising *Betula* (p=0.02), *P. sibirica* (p=0.001) and primary forest cover
(p=0.00), whereas that significantly increases with decreasing *Larix* (p=0.00), *Picea*
(p=0.001) abundance (Table 2, Fig. 1).
**4.2. Holocene climate-fuel feedbacks across the selected different sites**
**4.2.1. Southeastern/western Altai Mountains within the steppe zone (Region A):**
Multi-proxy records from four lacustrine systems (Achit Nuur, Tolbo, Alahake,
and Kuchuk Lakes) reveal consistent late-Holocene amplification of biomass burning
(Fig. 4b), with distinct peak intervals at ~1.2-~0.79 cal. kyr BP in Achit Nuur,
~1.20-~0.65 cal. kyr BP in Tolbo Lake, ~1.44-~1.02 cal. kyr BP in Alahake Lake and
pronounced charcoal flux doubling during the past two millennia in Kuchuk Lake.
Pollen spectra demonstrate ecosystem-specific fuel configurations: alpine steppe
dominated by *Artemisia*-Poaceae in Tolbo Lake, montane *P. sibirica* taiga in Achit
Nuur, lowland *Picea-Larix* mixed forest in Alahake Lake (Sun et al., 2013; Hu et al.,
2024; Li et al., 2021; Rudaya et al., 2020). The GAMs results show that the biomass
burning in Achit Nuur and Tolbo Lake is mainly controlled by the primary forest
cover. Among them, *Larix* (41.9%) and *P. sibirica* (34.5%) play a major role in the





biomass burning in Achit Nuur and Tolbo Lake, and *P. sibirica* (13.3%) plays a major
role in Tolbo Lake. The main sources of combustion in Alahake Lake are birch trees,
while those in Kuchuk Lake are *Betula* and *P. sylvestris* forest.
The early Holocene exhibited suppressed burning under moisture-limited
productivity (Zhang and Zhang, 2025). Precipitation from the mid-Holocene to ~2 cal
kyr BP (Hu et al., 2024; Zhang and Zhang, 2025) increases facilitated the expansion
of woody vegetation cover and fuel accumulation rates tripling (Sun et al., 2013).
Notably, after ~2 cal. kyr BP, anomalous biomass burning peaks recorded across four
archives likely correlate with agro-pastoral expansion markers (Cerealia pollen >5%)
and microcharcoal morphotype changes, signifying anthropogenic fire regimes
surpassing natural variability (Li et al., 2021; Xiao et al., 2022; Li et al., 2024;
Rudaya et al., 2020). The Tolbo Lake sequence preserves a deglacial signature (~11.5-
~10 cal. kyr BP) featuring charcoal peak preceding local vegetation establishment (Hu
et al., 2024), which is interpreted as pre-glacial reworking of Pleistocene-aged
charcoal during meltwater pulses (Blyakharchuk et al., 2024).
**4.2.2. West Siberian plain (Region B, n=4):**
Situated on the Ob' River low terrace (83 m asl), this pine (*P. sylvestris*)-birch
(*Betula*) dominated Rybnaya Mire exhibits the higher charcoal influx in the middle
Holocene with no big charcoal pulse during last 50 years (Feurdean et al., 2020) (Fig.
4c). GAM analysis reveals conifer-dominated fire controls: *Picea* cover explains
44.5% variance and *Betula* contributes 18.4% (Table 2). As part of the Great Vasygan
Mire (south taiga biome), vegetation in Plotnikovo mire (Fig. 4c) is dominated by
Scots pine (*P. sylvestris*) together with *Betula* and admixture of *Picea*. Biomass
burning curve has a quick increase since ~2 cal. kyr BP (Feurdean et al., 2020) with
the 39.7% deviance explained by primary forest cover (Table 2). Shchuchye Lake
demonstrates phased fire regime: strong charcoal pulse at ~12-~11 cal. kyr BP and
late-Holocene intensification (Fig. 4c). Key fire events in Ulukh- Chayakh mire
occurred in the last millennium and at ~4.5-~3 cal. kyr BP (Fig. 4c).
Cross-site synthesis of fire regimes in the west Siberian plain exhibits three
distinct fire phases. In details, burning pulse (only in Shchuchye Lake) at ~12-~11 cal.





kyr BP might be related with the meltwater-mediated charcoal deposition
(Blyakharchuk et al., 2024). The Pre-Holocene permafrost maintained waterlogged
soils, suppressing ignitions. With disappearance of permafrost soils became drier and
fires spread more easy when the time transited into the warming Holocene
(Blyakharchuk et al., 2024). The second higher biomass burning at ~8.5-~6 cal. kyr
BP was showed in Rybnaya peat, which is related with precipitation-driven higher
*Larix* pollen (Feurdean et al., 2022; Zhang and Zhang, 2025). The ~4.2 cal kyr BP
burning maximum across all sites coincides with regional megadrought conditions
(Feurdean et al., 2022) and emergent pastoralist fire use (Li et al., 2024). The GAMs
analysis reveal the divergent fire-vegetation relationships: (1) Negative correlation at
Rybnaya/Plotnikovo (canopy >75%): Reduced understory fuels and microclimatic
humidity limit fire spread; (2) Positive correlation at Shchuchye Lake (canopy <65%):
Open structure promotes flammable grass undergrowth.
**4.2.3. Northern Altai Mountains (Region C, n=4):**
Chudnoye Mire is situated in a remote mountain taiga near the upper limit of the
forest (Fig. 1). This region experienced a decline in biomass burning during the early
to mid-Holocene, followed by an intensification in the late Holocene (Fig. 4d).
Biomass burning can often explain the changes in dominant tree species within
mountain forests, particularly the positive correlation observed in *Larix* and *Picea*
pollen (Table 2). Tundra mire is characterized by dense forests of *Abies* and *Betula*, as
reflected in the pollen data. The charcoal influx exhibited a decreasing trend prior to
~4 cal. kyr BP, after which it began to increase. Mokhovoe Bog, which is covered by
birch forest-steppe, shows four peaks in charcoal influx at approximately ~11.5-~9.5
cal. kyr BP, ~8.5-~7 cal. kyr BP, ~5.6-~4 cal. kyr BP, and ~1.5-~1 cal. kyr BP. The
only statistical connection between *Picea* and biomass burning may be attributed to
increased bioproductivity of the landscape and the availability of fuel due to a more
humid climate. Kuatang Lake is located in dark coniferous wet mountain taiga, where
the charcoal influx has shown a clear increase since ~3.5 cal. kyr BP, followed by a
decreasing trend (Fig. 4d). The positive correlation between charcoal influx and
*Betula* pollen, contrasted with the negative correlations with *Abies, P. sibirica* and *P.*



*sylvestris*, suggests that the increased charcoal influx since ~3.5 cal. kyr BP may be attributed to the expansion of birch forest.

The regional synthesis of biomass burning reveals two distinct trends during the Holocene: a gradual decline in the early to mid-Holocene, followed by an increase in the late Holocene that subsequently exhibited a downward trajectory. Elevated charcoal influx in the early to mid-Holocene was predominantly recorded at Chudnoye Mire, Mokhovoe Bog and Tundra Mire. Notably, Mokhovoe Bog demonstrates a 2.1-fold higher charcoal influx compared to Chudnoye Mire and Tundra Mire, likely attributable to its ecotonal position within the forest-steppe transition zone, where progressive vegetation expansion during the early Holocene enhanced fuel availability. The increase in charcoal influx during the late Holocene, observed across all four sites, correlates with regional climatic humidification and intensified anthropogenic activities (Blyakharchuk et al., 2023; Li et al., 2024). Of particular significance, Mokhovoe Bog exhibits the most pronounced charcoal fluxes, reflecting persistent human occupation of these resource-rich landscapes since the Mesolithic era (Blyakharchuk, 2022).

**4.2.4. Central Altai Mountains within the forest zone (Region D, n=3):**

Holocene biomass burning exhibited an increasing trend in Kendegelukol Lake, Uzunkol Lake and Dzhangyskol Lake (Fig. 4e), with a notably pronounced expansion occurring in the late Holocene. A particularly strong increase in biomass burning was observed since ~1.2 cal. kyr BP in Uzunkol Lake and since ~0.5 cal. kyr BP in Dzhangyskol Lake. Notably, Uzunkol Lake recorded higher levels of biomass burning at ~9.5-~9 cal. kyr BP, coinciding with the transition from a dominant steppe landscape to a forest landscape; however, biomass burning did not maintain elevated levels following this transition (Blyakharchuk et al., 2004). The abnormal peak in charcoal influx at ~9.5-~9 cal. kyr BP was likely caused by an unstable fire regime during the onset of forested landscapes, which were particularly susceptible to ignition due to the prevailing dry climate and the increased availability of fuel from the spread of trees and shrubs (Blyakharchuk & Pupysheva, 2022). Following this transition, biomass burning at Uzunkol Lake decreased, indicating a shift to a more



stable climate and a reduction in fire frequency (Blyakharchuk et al., 2004). In
contrast, Kendegelukol Lake and Dzhangyskol Lake exhibited only a slight increase
in biomass burning, suggesting that Uzunkol Lake may be more sensitive to local fires
due to its location in the forest-steppe transition zone. This observation is supported
by similar research indicating that the forest-wooded grassland ecotone was highly
sensitive to climate variability during the Holocene (Lezine et al., 2023).

Despite minor variations in early Holocene biomass burning, these three lakes

demonstrate a statistically significant intensification of fire activity at ~4.5 cal. kyr BP,
with particularly pronounced amplification during the last millennium (Blyakharchuk
& Pupysheva, 2022). GAMs analysis reveals strong positive associations between
biomass burning and the pollen abundances of *Abies*, *Betula* and *P. sylvestris* across
these sites (Table 2), suggesting that fuel-load accumulation through late-Holocene
forest expansion drove shifts in fire regimes. The anomalous surge in biomass burning
post-1.0 cal. kyr BP likely reflects synergistic anthropogenic drivers, including
intensified pastoral burning practices and land clearance (Blyakharchuk et al., 2004,
2008). The regional synthesis demonstrates a sustained upward trajectory in biomass
burning throughout the Holocene, culminating in a 2.3-fold increase over the past two
millennia relative to early Holocene baselines.
**4.2.5. Central Altai Mountains above the forest limit (Region E, n=3):**

Regional integrated Z-scores indicate a consistent decline in biomass burning

prior to ~2 cal. kyr BP, followed by a rapid increase thereafter (Fig. 4f). GAMs reveal
that *Picea* in Tashkol Lake, *Picea* and *P. sylvestris* in Akkol Lake, and *Larix* and
*Picea* in Grusha Lake were the primary materials for biomass combustion (Table 2).
Significant differences in biomass burning were observed among the three lakes.

Tashkol Lake, situated above the modern forest limit at 2150 m a.s.l., was

covered by ice during the glacial period (Blyakharchuk et al., 2004). The sharp peak
in charcoal influx around ~11-~10.5 cal. kyr BP was likely caused by the redeposition
of microcharcoal by glacial meltwaters (Blyakharchuk et al., 2004). Subsequently, the
forested landscapes of central Altai between ~10.5 and ~4 cal. kyr BP, along with late
Holocene cooling, are clearly reflected in the biomass burning patterns of Tashkol



Lake, indicating the climate-dependent changes in bioproductivity and fuel
availability in high-elevation landscapes. The exceptionally high rate of charcoal
influx during the late glacial period, around 12-11 cal. kyr BP, can be attributed to the
allochthonous origin of redeposited old charcoal in Grusha Lake (Blyakharchuk et al.,
2004). Given that Grusha Lake is located at a high elevation (2413 m a.s.l.) and was
covered by glaciers during the glacial period (Rudoy and Yatsuk, 1986),
microcharcoal particles accumulated on the glacier surface throughout the glaciation.
As the glacier melted, these microcharcoal particles were washed into the lake basin
by meltwater. This hypothesis is supported by the very high rate of sediment
accumulation in Grusha Lake around ~12-~11 cal. kyr BP (Blyakharchuk et al., 2007).
Following deglaciation, the previously bare areas became vegetated, leading to a
sharp decrease in the redeposition of microcharcoal at ~10.5 cal. kyr BP. The overall
trend of charcoal influx in Akkol Lake is similar to that of Grusha Lake, with the
exception of the absence of a peak around ~12-~11 cal. kyr BP. This discrepancy can
be explained by the lower-elevation Akkol Lake, where glacial cover was absent,
resulting in drier conditions and a lack of redeposited microcharcoal following
deglaciation (Blyakharchuk et al., 2007).
**4.2.6. Western Sayan Mountains (Region F, n=3):**
Three peat cores―Lugovoe Peat, Bezrybnoye Mire and Buibinskoye Mire―
exhibited a decreasing trend in biomass burning throughout the Holocene (Fig. 4g). In
Buibinskoye Mire, a peak in biomass burning is observed around ~12-~11 cal. kyr BP.
During the late glacial and early Holocene, permafrost likely extended into the soils,
allowing only *Picea* to thrive in the Western Sayan (Blyakharchuk et al., 2022). As
permafrost receded, the prevalence of *Picea* diminished, giving way to *P. sibirica* and
*Abies*. Following the onset of forestation around ~11 cal. kyr BP, a sharp increase in
biomass burning occurred. However, prior to this, between ~11.5 and ~11 cal. kyr BP,
intense fires devastated spruce forests. Charcoal influx from three sites demonstrated
a similar trend of increase between ~10.5 and ~7 cal. kyr BP, followed by a gradual
decline in the late Holocene. The warmer climate during the Holocene climatic
optimum likely enhanced the bioproductivity of mountain forests in the Western



Sayan, resulting in increased fuel availability for fires (Blyakharchuk et al., 2013,
2022). The dominance of fire-avoiding *Abies* contributed to the elevated levels of
biomass burning during this period. With the onset of late Holocene cooling after ~7
cal. kyr BP, the rate of biomass burning decreased.
The three sites in the Western Sayan Mountains are situated between the upper
and lower limits of forest, leading to similar trends in the composition and content of
primary forest cover throughout the Holocene (Blyakharchuk et al., 2013, 2022). The
GAMs results indicate that *Abies* and *Larix* in Lugovoe Mire are the primary
contributors to biomass burning, while *Abies* in Buibinskoye Mire also plays a
significant role (Table 2). Although no significant relationship was found between
biomass burning and vegetation in Bezrybnoye Mire, the fire-resistant species *P.*
*sylvestris* (Feurdean et al., 2022) accounted for the largest deviance explanation
(28.10%) for biomass burning (Table 2). This suggests that the expansion of *P.*
*sylvestris* forests led to a reduction in the area of other combustible materials,
supported by the negative correlation between the spread of *P. sylvestris* and the
decrease in biomass combustion in Lugovoe Mire and Buibinskoye Mire (Table 2).
Consequently, the fire-resistant *P. sylvestris* can proliferate in the piedmonts of the
Western Sayan Mountains at the expense of fire-avoiding *Abies*. The dominance of
fire-resistant *P. sylvestris* has contributed to the reduction of biomass burning in the
forested areas of the Western Sayan.
**4.2.7. Khangai Mountains (Region G, n=3):**
Higher biomass burning was observed between ~3.5 and ~3.1 cal. kyr BP in Olgi
Lake (2012 m a.s.l.), between ~3.7 and ~3.3 cal. kyr BP in Shireet Naiman Nuur
(2429 m a.s.l.), and between ~2.4 and ~2.1 cal. kyr BP in Ugii Nuur (1330 m a.s.l.)
(Fig. 4h). Pollen data suggest that forest vegetation currently exists only at lower
elevations in the Khangai Mountains, while higher elevations remain devoid of forest
cover (Unkelbach et al., 2021; Barhoumi et al., 2024; Wang et al., 2011). Ugii Nuur
exhibited significantly higher biomass burning than both Olgi Lake and Shireet
Naiman Nuur, likely due to greater steppe vegetation coverage at lower elevations,
which provided abundant burning sources and stronger human influence around



~2.4-~2.1 cal. kyr BP (Wang et al., 2011). Although Shireet Naiman Nuur recorded a
gradual decline in biomass burning during the middle and late Holocene, its charcoal
influx was considerably lower than that of Olgi Lake and Ugii Nuur (Fig. 4h). The
charcoal data from high-elevation Shireet Naiman Nuur may reflect only climate-
induced decreases in biomass burning (late Holocene cooling), whereas the charcoal
data from Olgi Lake and Ugii Nuur indicate clear human influence around ~3.4-~3.1
cal. kyr BP and ~2.4-~2.1 cal. kyr BP, respectively. The GAMs analysis revealed that
biomass burning in Olgi Lake was negatively correlated with primary forest cover and
other woody types (Fig. S8), suggesting that biomass burning around Olgi Lake was
primarily controlled by herbaceous-dominated steppe vegetation. In contrast, biomass
burning in Shireet Naiman Nuur and Ugii Nuur was positively correlated with
primary forest cover and other woody types, indicating that biomass burning in these
areas was mainly regulated by woody vegetation, with *P. sibirica* having the highest
explanatory power (Table 2). According to pollen data, forests also existed at high
elevations near Shireet Naiman Nuur (2429 m a.s.l.) between ~7.5 and ~4 cal. kyr BP,
but did not grow near Olgi Lake (2012 m a.s.l.) (Unkelbach et al., 2021; Barhoumi et
al., 2024). During the middle Holocene optimum, some high-elevation areas of the
Khangai Mountains were covered by forests with high bioproductivity, which
contributed to increased biomass burning. However, the Khangai Mountains gradually
deforested during the late Holocene, leading to a decrease in biomass burning to
present low levels (Unkelbach et al., 2021; Barhoumi et al., 2024; Wang et al., 2011).
The role of *Picea* near Olgi Lake was more significant during the early Holocene
(~9.5-~8.5 cal. kyr BP), decreased during the period from ~8.5 to ~2 cal. kyr BP, and
then increased again after ~1.5 cal. kyr BP (Unkelbach et al., 2021; Barhoumi et al.,
2024). This fluctuation may be attributed to increased humidity, as *Picea* requires
wetter soils than *Pinus* (Blyakharchuk et al., 2013). The maximum charcoal influx
around ~3-~4 cal. kyr BP may be linked to early human influence (Xiang et al., 2023)
or to climatic shifts during the mid-Holocene transition (Zhao et al., 2017). This
climatic shift may have caused intense fires across all areas of the Khangai, resulting
in widespread deforestation. This hypothesis is supported by the decrease in the



contents of forest pollen in Shireet Naiman Nuur following the charcoal maximum
around ~3-~4 cal. kyr BP (Barhoumi et al., 2024).

**4.3. Holocene climate-fuel feedbacks across the different regions**

The relatively low biomass burning in the southeastern/western Altai Mountains
within the steppe zone prior to the last 2000 years coincided with low vegetation
cover (Sun et al., 2013; Hu et al., 2024; Li et al., 2021; Rudaya et al., 2020),
indicating that the drought-induced low vegetation cover inhibits fire occurrence
(Zhang et al., 2022). Since the last 2000 years, the rapid increase in biomass burning
has been attributed to changing climate conditions and intensified human activities
(Hu et al., 2024; Tian et al., 2021; Zhang and Zhang, 2025; Rudaya et al., 2020). A
similar pattern of low biomass burning prior to the last 2000 years was recorded in the
central Altai Mountains within the forest zone, including Kendegelukol, Uzunkol and
Dzhangyskol Lake. In Kendegelukol and Uzunkol Lake, forest components exceeding
70% suggest that dense forest coverage in the surrounding landscapes may limit
biomass burning (Carter et al., 2020). In Dzhangyskol Lake, situated in the
forest-steppe transition zone, the sustained low biomass burning before the last 2000
years may be attributed to lower vegetation productivity (Blyakharchuk et al., 2004,
2008). The significant increase in biomass burning over the past 2000 years across
these records may be directly related to intensified cattle grazing and human
settlement (Feurdean et al., 2020; Li et al., 2024; Rudaya et al., 2020; Xiang et al.,
2023; Zhang et al., 2022). Increased biomass burning around ~4.5-~3 cal. kyr BP may
be linked to human influence, as indicated by the presence of Triticum pollen
(Blyakharchuk et al., 2004, 2008). These findings are associated with the
development of ancient cultures (Blyakharchuk et al., 2004, 2008; Xiang et al., 2024).
In stark contrast to the trends observed in the first two regions, biomass burning
has shown an overall decline since the Holocene in the central Altai Mountains above
the forest limit, the western Sayan Mountains and the Khangai Mountains. The
gradual decrease in biomass burning above the timberline in the central Altai
Mountains is primarily influenced by the response of forest vegetation cover to
temperature changes. In the Western Sayan Mountains, the main forest vegetation



cover exceeds 80%, indicating that material availability is not a limiting factor for regional biomass burning. The GAMs analysis reveals that the decline in biomass burning in the Sayan Mountains is significantly associated with changes in forest composition. Specifically, the increase in Siberian pine and European larch since the Holocene has led to a significant decline in fir, birch, larch, and spruce components, resulting in a notable decrease in combustible materials at the three sites. Therefore, the decline in Holocene biomass in the Sayan region is primarily driven by changes in forest composition under temperature regulation. Notably, unlike the gradual increase in Holocene biomass burning observed in Kendegelukol and Uzunkol Lake, which are also in forested regions, there has been no overall decline in the Sayan region. This discrepancy is primarily attributed to human activities that have altered the occurrence of regional fires.

Although Holocene biomass burning in the Khangai Mountains exhibits an overall gradual decline, it can be categorized into two distinct phases: an increase over the past 2,000 years, followed by a gradual decline post-2000 year (Unkelbach et al., 2021; Barhoumi et al., 2024). The biomass burning characteristics during the earlier phase resemble those observed in the southeastern and western Altai Mountains, primarily due to increased humidity in the region, which led to a rise in combustible materials. In the later phase, despite the humid climate, the absence of a significant increase in biomass burning in the Khangai Mountains may be attributed to human grazing activities that have fragmented surface vegetation (Zhang S.J. et al., 2021). This assertion is supported by the studies of modern landscape, where livestock grazing eliminates most of the fuels necessary to sustain a fire (Umbanhowar et al., 2009; Zhang et al., 2022). The impact of human activities is also evident in areas above the timberline in the central Altai Mountains and the Sayan Mountains; however, the timing of this impact in the central Altai Mountains (~2.5 cal. kyr BP) predates that in the Sayan Mountains (~1 cal. kyr BP).

Biomass burning in the northern Altai Mountains demonstrates a gradual decline during the early to middle Holocene (Fig. 4d), a pattern consistent with trends observed above the upper forest line in the central Altai Mountains and the Sayan





Mountains (Fig. 4f-g). This early to mid-Holocene decline is likely related to
temperature-regulated forest vegetation dynamics. The late-Holocene increase in
biomass burning is associated with the intensified anthropogenic disturbances
(Blyakharchuk et al., 2024; Blyakharchuk, 2022). The West Siberian Plain exhibits
four peaks of Holocene biomass burning at ~12-~11 cal. kyr BP, ~8.4-~6.6 cal. kyr BP,
~4.4-~4.2 cal. kyr BP and ~1.4 cal. kyr BP (Fig. 2c). The first peak recorded at
Shchuchye Lake derived from ancient sediments in formerly glaciated or permafrost-
affected areas (Blyakharchuk et al., 2024). The second peak at Rybnaya Peat
corresponds to high *Larix* coverage around the mire (Feurdean et al., 2022). The third
peak is supported by biomass burning from the southeastern and western Altai, central
Altai, Sayan and Khangai Mountains, potentially linked to regional aridity or
increased human activity (Zhang and Zhang, 2025). Notably, a Bronze Age charcoal
pulse (~4-~3 cal. kyr BP) at Kuatang Bog and an Early Iron Age pulse (~3 cal. kyr BP)
at Tundra Mire coincide with the Kuznetski Alatau Mountains—a known center of
ancient Siberian metallurgy (Slavnin and Sherstova, 1999). The fourth peak directly
corresponds to numerous archaeological sites of ancient human cultures, indicating
densely populated areas (Panyushkina, 2012; Agatova et al., 2014; Xiang et al., 2024).
**5. Conclusions**
This study presents a long-term fire record from the steppe zone of Western
Mongolia and evaluates the spatial variations in biomass burning and its relationship
with forest composition across the Altai-Sayan Mountains and adjacent plains. Our
findings indicate that the reduction in biomass burning during the Holocene can be
attributed to the temperature-regulated woody biomass above the forest limit in the
central Altai Mountains, as well as a decrease in combustible components (*Larix*,
*Abies* and *Picea*) in the western Sayan Mountains and northern Altai Mountains.
Global cooling and increased moisture during the late Holocene contributed to the
declining trend of biomass burning in the western Sayan Mountains and central Altai
Mountains within the forest zone. A notable increase in biomass burning since ~4 cal.
kyr BP has been observed in areas historically populated by various archaeological
cultures. The late Holocene rise in biomass burning occurred in regions that were



previously less susceptible to fires due to either excessively wet and cool climates
(such as the plains and mountain taiga of Western Siberia and the Altai Mountains) or
excessively dry climates with sparse steppe or desert-steppe vegetation that could not
provide sufficient fuel for fires. The latter scenario is characteristic of southeastern
Altai, particularly in the steppe areas surrounding Kuchuk Lake, as well as in Uzunkol
and Dzhangyskol lakes located in intermountain hollows covered by steppe vegetation.
Intensified human activities, including agriculture and pasture, have led to increased
fire frequency since ~2 cal. kyr BP in the southeastern Altai Mountains, the West
Siberian Plain, and the forest zone of the middle Altai Mountains. Conversely, the
significant decline in biomass burning in the Khangai Mountains may be attributed to
vegetation fragmentation caused by grazing activities. This research elucidates the
long-term relationship between biomass burning and forest composition/density
across different vegetation zones in the Altai-Sayan Mountains and adjacent plains,
which holds practical significance for predicting and managing future fire dynamics.
**CRediT authorship contribution statement**
Dongliang Zhang: Writing – review & editing, Validation, Methodology, Funding
acquisition, Conceptualization. Blyakharchuk Tatiana, Aizhi Sun, Xiaozhong Huang:
Writing – original draft, Visualization, Methodology, Data curation. Yuejing Li – Data
curation.
**Declaration of Competing Interest**
The authors declare that they have no known competing financial interests or personal
relationships that could have appeared to influence the work reported in this paper.
**Acknowledgment.** This research was financially supported by National Natural
Science Grants of China (No. 42471183), Youth Innovation Promotion Association of
Chinese Academy of Sciences (No. 2022447) and National Natural Science Grants of
China (No. 42220104001). We thank anonymous reviewers for their valuable
comments, which significantly improved the manuscript.

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

Preliminary study on pollen, charcoal records and environmental evolution of

Alahake Saline Lake in Xinjiang since 4,700 cal yr BP. Quat. Int. 513, 8–17.

Li, Y., Zhang, D., Zhang, Y., Sun, A., Li, X., Huang, X., Zhang, Y., Li, Y., 2024.

Distentanging the late-Holocene human–environment interactions in the Altai

Mountains within the Arid Central Asia. Palaeogeography, Palaeoclimatology,



Palaeoecology, 654, 112466.

Liu, F., Liu, H., Xu, C., Shi, L., Zhu, X., Qi, Y., He, W., 2021. Old-growth forests

show low canopy resilience to droughts at the southern edge of the taiga. Global

Change Biology, 27(11), 2392-2402.

Lung, T., Lavalle, C., Hiederer, R., Dosio, A., Bouwer, L.M., 2013. A multi-hazard

regional level impact assessment for Europe combining indicators of climatic

and non-climatic change. Global and Environmental Change, 23, 522-536.

Moritz, M.A., Batllori, E., Bradstock, R.A., Gill, A.M., Handmer, J., Hessburg, P.F.,

Leonard, J., McCaffrey, S., Odion, D.C., Schonennagel, T., Syphard, A.D., 2014.

Learning to coexist with wildfire. Nature, 515(7525), 58-66.

Panyushkina, I.P., 2012. Climate-Induced changes in Population Dynamics of

Siberian Scythians (700-250 B.C.). Climate, Landscapes and Civilizations.

Geographycal Monograph Series 198, 145-154.

Pitkanen, A., Huttunen, P., Tolonen, K., Jungner, H., 2003. Long term fire frequency

in the spruce-dominated forests of the Ulvinsalo strict nature reserve. Finland.

Forest Ecology and Management, 176(1-3), 305-319.

Ponomarev, E.I., Kharuk, V.I., 2016. Wildfire occurrence in forests of the Altai-Sayan

region under current climate changes. Contemporary Problems of Ecology, 9,

29-36.

Power, M.J., Marlon, J., Ortiz, N., et al., 2007. Changes in fire regimes since the Last

Glacial Maximum: an assessment based on a global synthesis and analysis of

charcoal data. Climate dynamics, 30, 887-907.

Pupycheva, M.A., Blyakharchuk, T.A., 2024. Late Holocene dynamics of fires in the

forest-steppe zone (a case study of the Nikolaevsky Ryam) Geografiya I

prirosnye resursy. 1, 54-61 (in Russian).

Reimer, P.J., Austin, W.E., Bard, E., Bayliss, A., Blackwell, P.G., Ramsey, C.B.,

Talamo, S., 2020. The IntCal20 Northern Hemisphere radiocarbon age

calibration curve (0–55 cal kBP). Radiocarbon 62 (4), 725–757.

Rudaya, N., Sergey, K., Michał, S., Xianyong, C., Snezhana, Z., 2020. Postglacial

history of the steppe Altai: climate, fire and plant diversity. Quat. Sci. Rev. 249,

106616.



Rudoy, A.N., Yatsuk, T.Yu., 1986. The palaeogeography of southeastern Altai.
Chetvertichnaya geologiya i pervobytnaya arkheologiya. Thesis of conference,
Ulan-Ude, 73–75.
Shi, C.M., Liang, Y., Gao, C., Wang, Q.H., Shu, L.F., 2020. Drought-modulated
boreal forest fire occurrence and linkage with La Nina events in Altai Mountains
Northwest China. Atmosphere, 11, 956.
Sun, A., Feng, Z.D., Ran, M., Zhang, C.J., 2013. Pollen-recorded bioclimatic
variations of the last ~22,600 years retrieved from Achit Nuur core in the western
Mongolian Plateau. Quat. Int. 311, 36-43.
Shi, C.M., Liang, Y., Gao, C., Wang, Q.H., Shu, L.F., 2020. Drought-modulated
boreal forest fire occurrence and linkage with La Nina events in Altai Mountains
Northwest China. Atmosphere, 11, 956.
Shumilova, L.V., 1962. Botanical Geography of Siberia. Tomsk University Press:
Tomsk. (in Russian).
Slavnin, V.D., Sherstova L.I. 1999. Aerchaeologic-Ethnographic Essay of Northern
Khakassia in the Area of Geological Polygon of Siberian High School). Tomsl
Polytechnical University Press, Tomsk (in Russian)
Stockmarr, J.A., 1971. Tablets with spores used in absolute pollen analysis. Pollen
spores, 13, 61–621.
Tang, G., Yang, S., Miao, Y., et al., 2022. Grain size characteristics of microfossil
charcoal and the environmental implications in loess deposits from Ganzi,
Western Sichuan Plateau. Journal of Lanzhou University (Natural Sciences) 58
(03), 298–305 (in Chinese with English astract).
Stockmarr J.A., 1971. Tablets with spores used in absolute pollen analysis. Pollen
spores, 13, 61-621.
Umbanhowar Jr, C.E., Shinneman, A.L., Tserenkhand, G., Hammon, E.R., Lor, P.,
Nail, K., 2009. Regional fire history based on charcoal analysis of sediments
from nine lakes in western Mongolia. Holocene 19(4), 611-624.
Unkelbach, J., Dulamsuren, C., Klinge, M., Behling, H., 2021. Holocene high-
resolution forest-steppe and environmental dynamics in the Tarvagatai
Mountains, northcentral Mongolia, over the last 9570 cal yr BP. Quat. Sci. Rev.



266, 107076.

Walker, X.J., Baltzer, J.L., Cumming, S.G., Day, N.J., Ebert, C., Goetz, S., Johnstone,
778        J.F., Potter, S., Rogers, B.M., Schuur, E.A.G., Turetsky, M.R., Mack, M.C., 2019.
Increasing wildfires threaten historic carbon sink of boreal forest soils. Nature,
572, 520-523.

Wang, W., Ma, Y.Z., Feng, Z.D., Narantsetseg, Ts, Liu, K.B., Zhai, X.W., 2011. A
prolonged dry mid-Holocene climate revealed by pollen and diatom records from
Lake Ugii Nuur in central Mongolia. Quat. Int. 229 (1e2), 74-83.

Wang, Z., Miao, Y., Zhao, Y., Zhang, Z., Zou, Y., Zhang, T., 2024. Preliminary
exploration of the fire activity recorded by microcharcoal in surface sediments of
Central and Western China. Quat. Sci. 44 (1), 201-213 (in Chinese with English
astract).

Wood, S.N., 2017. Generalized Additive Models: An Introduction with R (2nd
Edition). Chapman and Hall/CRC, pp1-476.

Xiao, Y., Xiang, L., Huang, X., et al., 2021. Moisture changes in the Northern
Xinjiang Basin over the past 2400 years as Documented in Pollen Records of Jili
Lake. Front. Earth Sci. 9, 741992.

Xiang, L., Huang, X., Sun, M., Panizzo, V. N., Huang, C., Zheng, M., Chen, F., 2023.
Prehistoric population expansion in Central Asia promoted by the Altai Holocene
climatic optimum. Nature Communications, 14(1), 3102.

Xinjiang Comprehensive Expedition Team, Institute of Botany, Chinese Academy of
Sciences, 1978. Vegetation and its utilization in Xinjiang. Beijing: Science Press.

Zhang, D.L., Feng, Z.D., 2018. Holocene climate variations in the Altai Mountains
and the surrounding areas: a synthesis of pollen records. Earth Sci. Rev. 185,
847-869.

Zhang, D., Huang, X., Liu, Q., Chen, X., Feng, Z., 2022. Holocene fire records and
their drivers in the westerlies-dominated Central Asia. Sci. Total Environ. 833,
155153.

Zhang, S.J., Lu, Y., Wei, W., Qiu, M., Dong, G., Liu, X., 2021. Human activities have
altered fire-climate relations in arid Central Asia since ~1000 a BP: evidence
from a 4200-year-old sedimentary archive. Sci. Bull. 66(8), 761-764.



Zhang, Y.Y., Feng, Z.D., 2024. Pollen-based quantitative reconstructions of Holocene

climate at Gun Nuur in the northern Mongolian Plateau. Palaeogeography,

Palaeoclimatology, Palaeoecology, 638, 112028.

Zhang, Y.Y., Zhang, D.L., 2025. Spatiotemporal patterns of pollen-based Holocene

precipitation variations in the Altai Mountains and the surrounding areas. Global

and Planetary Change, 251, 104832.

Zhao, Y., Liu, Y.L., Guo, Z.T., Fang, K.Y., Li, Q., Cao, X.Y., 2017. Abrupt vegetation

shifts caused by gradual climate changes in central Asia during the Holocene. Sci.

China Earth Sci. doi: 10.1007/s11430-017-9047-7




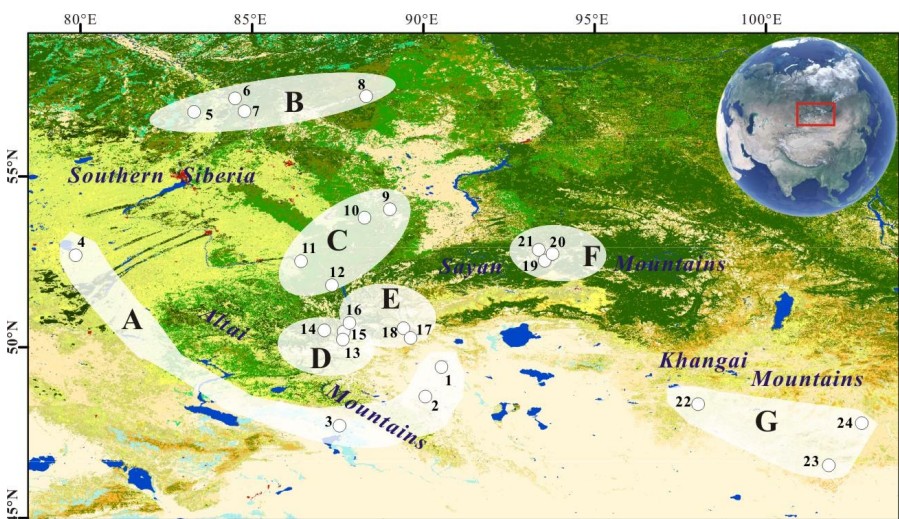

**Fig. 1.** Spatial distributions of the selected fossil pollen/charcoal sequences around the Altai-Sayan Mountains and adjacent plains. **Region A**: Achit Nuur (1), Tolbo Lake (2), Alahake Lake (3) and Kuchuk Lake (4); **Region B**: Rybnaya Mire (5), Plotnikovo Mire (6), Shchuchye Lake (7) and Ulukh–Chayakh Mire (8); **Region C**: Chudnoye Mire (9), Tundra Mire (10), Mokhovoe Bog (11) and Kuatang Mire (12); **Region D**: Dzhangyskol Lake (13), Uzunkol Lake (14) and Kendegelukol Lake (15); **Region E**: Tashkol Lake (16), Akkol Lake (17) and Grusha Lake (18); **Region F**: Buibinskoye Mire (19), Bezrybnoye Mire (20) and Lugovoe Peat (21); **Region G**: Olgi Lake (OL3) (22), Shireet Naiman Nuur (23) and Uggi Nuur (24).

.




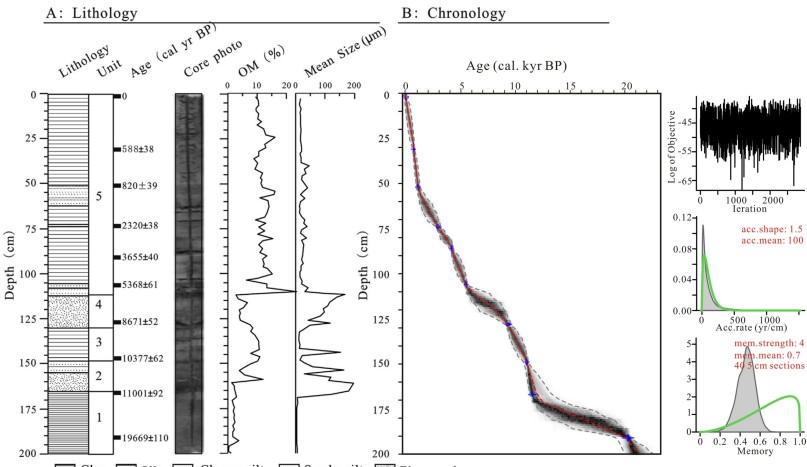

**Fig. 2.** Lithology, core photo, organic matter (OM), mean grain size and depth-age model in Achit

Nuur (modified from Sun et al., 2023).

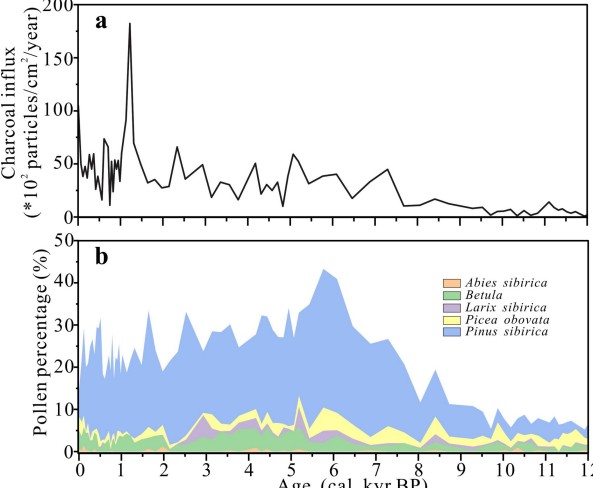

**Fig. 3.** Achit Nuur: biomass burning indicated by charcoal influx (a), vegetation change (b) (Sun

et al., 2013; this study).

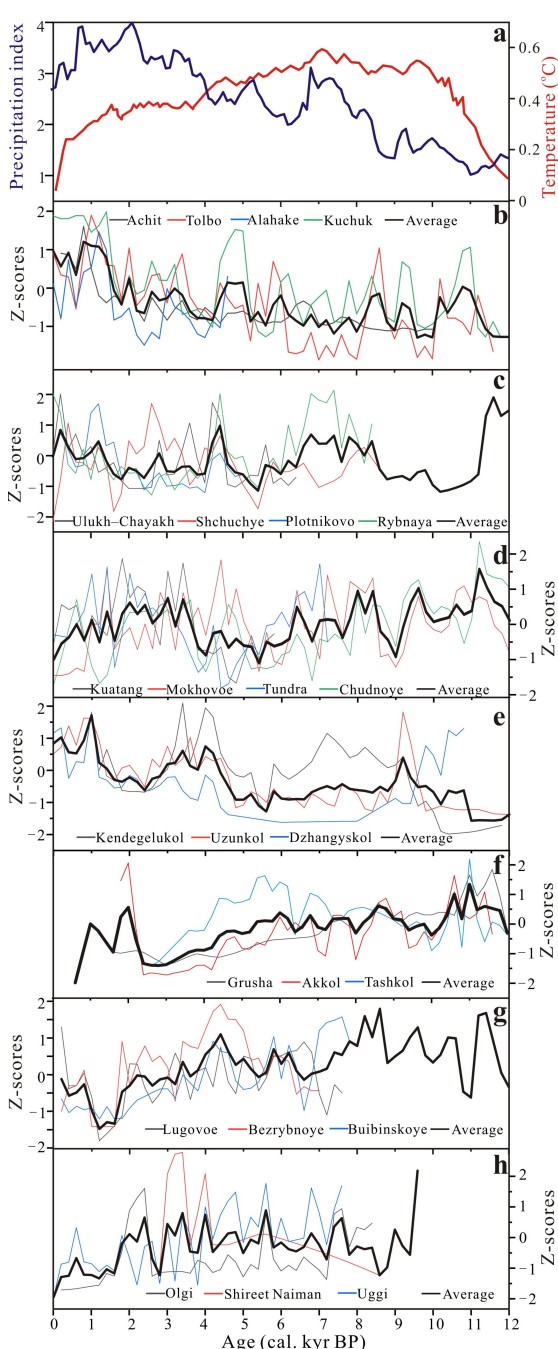

**Fig. 4.** Regional integrated biomass burning (b-g) under the context of temperature (Marcott et al., 2013) and precipitation index (a) in the Holocene interval (Zhang and Feng, 2018).







**Table 1** Detailed information of the selected sites around the Altai-Sayan Mountains and adjacent
plains.

| Region | No. | Site Name | Lat. (N) | Long. (E) | Elev. (m a.s.l.) | References |
|--------|-----|-----------|----------|-----------|------------------|------------|
| A | 1 | Achit Nuur | 49.42 | 90.52 | 1444 | Sun et al., 2013; this study |
|  | 2 | Tolbo Lake | 48.55 | 90.05 | 2080 | Hu et al., 2024 |
|  | 3 | Alahake Lake | 47.69 | 87.54 | 483 | Li et al., 2021 |
|  | 4 | Kuchuk Lake | 52.69 | 79.84 | 98 | Rudaya et al., 2020 |
| B | 5 | Rybnaya Mire | 57.28 | 84.49 | - | Feurdean et al., 2022 |
|  | 6 | Plotnikovo Mire | 56.88 | 83.30 | 120 | Feurdean et al., 2020 |
|  | 7 | Shchuchye Lake | 57.13 | 84.61 | 80 | Blyakharchuk et al., 2024 |
|  | 8 | Ulukh–Chayakh Mire | 57.34 | 88.32 | - | Feurdean et al., 2022 |
| C | 9 | Chudnoye Mire | 54.03 | 89.01 | 1147 | Blyakharchuk et al., 2024 |
|  | 10 | Tundra Mire | 53.79 | 88.27 | 247 | Blyakharchuk et al., 2024 |
|  | 11 | Mokhovoe Bog | 52.52 | 86.42 | 283 | Blyakharchuk, 2022 |
|  | 12 | Kuatang Mire | 51.81 | 87.32 | 650 | Blyakharchuk et al., 2024 |
| D | 13 | Dzhangyskol Lake | 50.18 | 87.73 | 1800 | Blyakharchuk et al., 2008 |
|  | 14 | Uzunkol Lake | 50.48 | 87.1 | 1985 | Blyakharchuk et al., 2004 |
|  | 15 | Kendegelukol Lake | 50.50 | 87.63 | 2050 | Blyakharchuk et al., 2004 |
| E | 16 | Tashkol Lake | 50.45 | 87.67 | 2150 | Blyakharchuk et al., 2004 |
|  | 17 | Akkol Lake | 50.25 | 89.62 | 2204 | Blyakharchuk et al., 2007 |
|  | 18 | Grusha Lake | 50.38 | 89.42 | 2413 | Blyakharchuk et al., 2007 |
| F | 19 | Buibinskoye Mire | 52.84 | 93.52 | 1377 | Blyakharchuk et al., 2022 |
|  | 20 | Bezrybnoye Mire | 52.81 | 93.50 | 1395 | Blyakharchuk et al., 2022 |
|  | 21 | Lugovoe Peat | 52.85 | 93.35 | 1299 | Blyakharchuk et al., 2013 |
| G | 22 | Olgi Lake(OL3) | 48.32 | 98.01 | 2012 | Unkelbach et al., 2021 |
|  | 23 | Shireet Naiman Nuur | 46.53 | 101.82 | 2429 | Barhoumi et al., 2024 |
|  | 24 | Uggi Nuur | 47.77 | 102.78 | 1330 | Wang et al., 2011 |




**Table 2** Correlation between the independent variables represented by pollen percentages (*Betula*, *Larix*, *Picea*, *Pinus sibirica*, *Pinus sylvestris* and primary forest cover (i.e., the summed percentage values of *Betula*, *Larix*, *Picea* and *Pinus*)) and the dependent variable (biomass burning; charcoal influx). The significance of each parameter is given by p values where \*\*\*p < 0.001; \*\*p < 0.01; \*p < 0.05.

| Site Name | Independent variable | edf | ref.df | F value | p-value | Deviance explained |
|---|---|---|---|---|---|---|
| Achit Nuur | *Abies* | - | - | - | - | - |
| | *Betula* | 2.75 | 3.47 | 3.40 | 0.02* | 21% |
| | *Larix* | 3.51 | 4.21 | 8.72 | 0.00*** | 41.9% |
| | *Picea* | 1 | 1 | 11.36 | 0.001** | 19.2% |
| | *Pinus sibirica* | 2.73 | 3.41 | 5.70 | 0.001** | 34.5% |
| | *Pinus sylvestris* | - | - | - | - | - |
| | Primary cover | 2.92 | 3.69 | 8.02 | 0.00*** | 41.5% |
| Tolbo Lake | *Abies* | - | - | - | - | - |
| | *Betula* | 6.96 | 8.01 | 1.76 | 0.09 | 7.04% |
| | *Larix* | 1.03 | 1.07 | 0.03 | 0.95 | 0.03% |
| | *Picea* | 2.97 | 3.75 | 4.47 | 0.002** | 8.11% |
| | *Pinus sibirica* | 2.68 | 3.39 | 9.55 | 0.00*** | 13.3% |
| | *Pinus sylvestris* | - | - | - | - | - |
| | Primary cover | 2.98 | 3.75 | 8.96 | 0.00*** | 14.3% |
| Alahake Lake | *Abies* | 1 | 1 | 0.57 | 0.45 | 1.1% |
| | *Betula* | 1 | 1 | 4.19 | 0.04* | 5.2% |
| | *Larix* | 6.85 | 7.94 | 1.42 | 0.19 | 11.6% |
| | *Picea* | 3.84 | 4.77 | 1.96 | 0.09 | 10% |
| | *Pinus sibirica* | 5.59 | 6.77 | 1.85 | 0.09 | 13% |
| | *Pinus sylvestris* | - | - | - | - | - |
| | Primary cover | 2.17 | 2.77 | 1.24 | 0.26 | 5.07% |
| Kuchuk Lake | *Abies* | 1.21 | 1.40 | 3.80 | 0.03* | 9.81% |
| | *Betula* | 1.38 | 1.67 | 16.18 | 0.00*** | 25.2% |
| | *Larix* | 1.11 | 1.21 | 0.01 | 0.98 | 0.19% |
| | *Picea* | 1.16 | 1.30 | 1.31 | 0.30 | 2.29% |
| | *Pinus sibirica* | 5.84 | 6.89 | 1.06 | 0.39 | 9.51% |
| | *Pinus sylvestris* | 6.54 | 7.64 | 2.61 | 0.01* | 25.5% |
| | Primary cover | 3.59 | 4.47 | 1.22 | 0.28 | 11% |
| Rybnaya Mire | *Abies* | 5.28 | 6.31 | 1.99 | 0.07 | 11.7% |
| | *Betula* | 4.90 | 6.00 | 3.32 | 0.004** | 18.4% |
| | *Larix* | 7.07 | 8.11 | 1.95 | 0.07 | 20.6% |
| | *Picea* | 8.15 | 8.79 | 14.1 | 0.00*** | 44.5% |
| | *Pinus sibirica* | 6.74 | 7.86 | 1.68 | 0.12 | 16.6% |
| | *Pinus sylvestris* | 2.03 | 2.54 | 1.06 | 0.35 | 4% |
| | Primary cover | 7.00 | 8.10 | 3.06 | 0.003** | 16.2% |



| | | | | | | |
|---|---|---|---|---|---|---|
| Plotnikovo Mire | *Abies* | 3.12 | 3.88 | 0.70 | 0.55 | 16.7% |
| | *Betula* | 2.69 | 3.36 | 1.40 | 0.26 | 19.6% |
| | *Larix* | 1 | 1 | 4.09 | 0.06 | 20.1% |
| | *Picea* | 2.12 | 2.65 | 1.54 | 0.26 | 15.1% |
| | *Pinus sibirica* | 1.68 | 2.11 | 0.41 | 0.7 | 4.85% |
| | *Pinus sylvestris* | 2.01 | 2.53 | 1.50 | 0.23 | 14.7% |
| | Primary cover | 4.43 | 5.21 | 4.07 | 0.004** | 39.7% |
| Schuchye Lake | *Abies* | 4.78 | 5.85 | 5.39 | 0.00*** | 37.4% |
| | *Betula* | 1 | 1 | 5.29 | 0.03* | 10.8% |
| | *Larix* | 1 | 1 | 63.71 | 0.00*** | 45.4% |
| | *Picea* | 2.19 | 2.71 | 3.77 | 0.02* | 17.5% |
| | *Pinus sibirica* | 1 | 1 | 27.6 | 0.00*** | 30.8% |
| | *Pinus sylvestris* | 3.15 | 3.90 | 3.31 | 0.02* | 21.2% |
| | Primary cover | 2.10 | 2.52 | 7.91 | 0.00*** | 24.7% |
| Ulukh–Chayakh Mire | *Abies* | 6.38 | 7.52 | 1.60 | 0.18 | 29.4% |
| | *Betula* | 1 | 1 | 6.44 | 0.01* | 13.4% |
| | *Larix* | 2.54 | 3.16 | 2.46 | 0.07 | 17.5% |
| | *Picea* | 2.45 | 3.12 | 1.46 | 0.23 | 16.7% |
| | *Pinus sibirica* | 1 | 1 | 0.66 | 0.42 | 1.82% |
| | *Pinus sylvestris* | 1 | 1 | 4.43 | 0.04* | 10.3% |
| | Primary cover | 4.26 | 5.08 | 1.46 | 0.22 | 16.9% |
| Chudnoye Lake | *Abies* | 1.75 | 2.17 | 2.09 | 0.14 | 8.52% |
| | *Betula* | 1.23 | 1.42 | 10.54 | 0.001** | 23.5% |
| | *Larix* | 2.06 | 2.57 | 3.84 | 0.03* | 14.7% |
| | *Picea* | 1.99 | 2.44 | 11.76 | 0.00*** | 30.3% |
| | *Pinus sibirica* | 4.33 | 5.25 | 3.38 | 0.01* | 26.6% |
| | *Pinus sylvestris* | 1 | 1 | 6.59 | 0.01* | 11.6% |
| | Primary cover | 1 | 1 | 1.97 | 0.17 | 3.5% |
| Tundra Mire | *Abies* | 2.16 | 2.75 | 0.78 | 0.57 | 3.83% |
| | *Betula* | 1 | 1 | 3.27 | 0.07 | 4.44% |
| | *Larix* | 6.41 | 7.35 | 4.32 | 0.00*** | 22.7% |
| | *Picea* | 1 | 1 | 0.09 | 0.77 | 0.13% |
| | *Pinus sibirica* | 2.39 | 2.99 | 0.83 | 0.46 | 4.66% |
| | *Pinus sylvestris* | 3.03 | 3.78 | 0.79 | 0.49 | 5.83% |
| | Primary cover | 1 | 1 | 2.79 | 0.10 | 3.53% |
| Mokhove Bog | *Abies* | 1.83 | 2.31 | 1.12 | 0.38 | 3.65% |
| | *Betula* | 6.81 | 7.88 | 2.07 | 0.05 | 17.2% |
| | *Larix* | 1.09 | 1.17 | 0.24 | 0.63 | 0.59% |
| | *Picea* | 2.59 | 3.22 | 3.54 | 0.02* | 11.9% |
| | *Pinus sibirica* | 1 | 1 | 0.00 | 0.96 | 0.003% |
| | *Pinus sylvestris* | 4.46 | 5.49 | 1.78 | 0.11 | 13% |
| | Primary cover | 5.04 | 6.19 | 0.91 | 0.48 | 10.3% |
| Kuatang | *Abies* | 2.45 | 3.14 | 2.78 | 0.04* | 13.8% |





| | | | | | |
|---|---|---|---|---|---|
| Mire | *Betula* | 1 | 1 | 29.13 | 0.00*** | 24.5% |
| | *Larix* | 1 | 1.00 | 0.06 | 0.81 | 0.08% |
| | *Picea* | 6.72 | 7.79 | 1.19 | 0.31 | 13.4% |
| | *Pinus sibirica* | 1.43 | 1.74 | 2.92 | 0.05* | 6.90% |
| | *Pinus sylvestris* | 1 | 1 | 5.83 | 0.02* | 6.51% |
| | Primary cover | 1 | 1 | 9.24 | 0.003** | 10.9% |
| | *Abies* | 3.64 | 4.53 | 0.45 | 0.79 | 16.9% |
| | *Betula* | 1.79 | 2.23 | 0.37 | 0.77 | 7.12% |
| Dzhangysk ol Lake | *Larix* | 1 | 1 | 0.05 | 0.83 | 0.33% |
| | *Picea* | 3.92 | 4.80 | 0.82 | 0.51 | 24.8% |
| | *Pinus sibirica* | 1.70 | 2.12 | 0.35 | 0.73 | 7.06% |
| | *Pinus sylvestris* | 3.05 | 3.75 | 1.22 | 0.29 | 22.8% |
| | Primary cover | 2.39 | 3.04 | 0.67 | 0.58 | 15.6% |
| | *Abies* | 1 | 1 | 5.329 | 0.02* | 7.04% |
| Uzunkol | *Betula* | 4.92 | 5.99 | 3.22 | 0.01** | 29.4% |
| | *Larix* | 1 | 1 | 14.38 | 0.00*** | 22.1% |
| Lake | *Picea* | 5.99 | 7.12 | 5.03 | 0.00*** | 40.1% |
| | *Pinus sibirica* | 2.04 | 2.57 | 1.99 | 0.14 | 14.7% |
| | *Pinus sylvestris* | 4.79 | 5.81 | 2.85 | 0.02* | 29.3% |
| | Primary cover | 2.17 | 2.69 | 1.39 | 0.27 | 14.2% |
| | *Abies* | 4.93 | 5.97 | 2.63 | 0.04* | 41.4% |
| | *Betula* | 5.87 | 7.04 | 2.78 | 0.02* | 49.4% |
| Kendegelu kol Lake | *Larix* | 1 | 1 | 3.11 | 0.09 | 9.63% |
| | *Picea* | 2.99 | 3.73 | 2.19 | 0.08 | 29.4% |
| | *Pinus sibirica* | 2.25 | 2.78 | 2.26 | 0.09 | 28.9% |
| | *Pinus sylvestris* | 1 | 1 | 18.48 | 0.00*** | 40% |
| | Primary cover | 1.57 | 1.91 | 3.58 | 0.06 | 26.9% |
| | *Abies* | 1 | 1 | 0.02 | 0.90 | 0.09% |
| | *Betula* | 1 | 1 | 0.08 | 0.79 | 0.36% |
| Tashkol | *Larix* | 1.56 | 1.92 | 0.20 | 0.82 | 3.52% |
| Lake | *Picea* | 6.69 | 7.81 | 2.35 | 0.04* | 40.7% |
| | *Pinus sibirica* | 1 | 1 | 0.004 | 0.95 | 0.02% |
| | *Pinus sylvestris* | 1 | 1 | 0.02 | 0.89 | 0.09% |
| | Primary cover | 3.00 | 3.75 | 0.90 | 0.48 | 17% |
| | *Abies* | 1.76 | 2.11 | 0.79 | 0.43 | 4.83% |
| | *Betula* | 1 | 1 | 0.96 | 0.33 | 1.76% |
| Akkol | *Larix* | 6.53 | 7.59 | 1.94 | 0.08 | 30.4% |
| Lake | *Picea* | 2.41 | 3.03 | 6.77 | 0.00*** | 31.6% |
| | *Pinus sibirica* | 4.35 | 5.41 | 1.90 | 0.1 | 23% |
| | *Pinus sylvestris* | 1 | 1 | 10.12 | 0.002** | 18.9% |
| | Primary cover | 8.47 | 8.92 | 5.49 | 0.00*** | 55.1% |
| Grusha | *Abies* | 1 | 1 | 0.62 | 0.44 | 2.75% |





| | | | | | |
|---|---|---|---|---|---|
| Lake | *Betula* | 1 | 1 | 0.88 | 0.36 | 3.93% |
| | *Larix* | 3.81 | 4.58 | 3.44 | 0.02* | 49.3% |
| | *Picea* | 2.18 | 2.71 | 3.30 | 0.05* | 35.80% |
| | *Pinus sibirica* | 1 | 1 | 0.60 | 0.45 | 2.67% |
| | *Pinus sylvestris* | 1.39 | 1.66 | 0.19 | 0.76 | 4.67% |
| | Primary cover | 2.55 | 3.18 | 12.7 | 0.00*** | 71.1% |
| Bezrybnoe Mire | *Abies* | 1.15 | 1.29 | 0.31 | 0.75 | 1.16% |
| | *Betula* | 1.74 | 2.20 | 1.63 | 0.22 | 8.85% |
| | *Larix* | 2.58 | 3.14 | 0.32 | 0.79 | 4.76% |
| | *Picea* | 1 | 1 | 2.13 | 0.15 | 4.49% |
| | *Pinus sibirica* | 1.37 | 1.66 | 0.39 | 0.75 | 2.18% |
| | *Pinus sylvestris* | 6.47 | 7.53 | 1.69 | 0.13 | 28.1% |
| | Primary cover | 1 | 1 | 0.01 | 0.93 | 0.02% |
| Buibinskoye Mire | *Abies* | 2.71 | 3.39 | 4.85 | 0.004** | 29.6% |
| | *Betula* | 2.11 | 2.69 | 2.29 | 0.10 | 17.4% |
| | *Larix* | 1 | 1 | 1.16 | 0.29 | 2.83% |
| | *Picea* | 1.52 | 1.87 | 0.71 | 0.40 | 4.85% |
| | *Pinus sibirica* | 2.02 | 2.57 | 2.70 | 0.05 | 17.4% |
| | *Pinus sylvestris* | 1 | 1 | 3.78 | 0.06 | 7.42% |
| | Primary cover | 3.61 | 4.42 | 2.47 | 0.06 | 22.6% |
| Lugovoe Mire | *Abies* | 1 | 1 | 6.32 | 0.02* | 15.3% |
| | *Betula* | 1 | 1 | 0.23 | 0.64 | 0.79% |
| | *Larix* | 5.00 | 5.91 | 3.89 | 0.01** | 43.5% |
| | *Picea* | 4.00 | 4.95 | 2.41 | 0.07 | 35.8% |
| | *Pinus sibirica* | 3.43 | 4.28 | 2.20 | 0.09 | 31% |
| | *Pinus sylvestris* | 8.81 | 8.98 | 3.21 | 0.01* | 60.5% |
| | Primary cover | 1.14 | 1.27 | 0.20 | 0.67 | 2.29% |
| Olgi Lake | *Abies* | - | - | - | - | - |
| | *Betula* | 4.89 | 5.96 | 2.91 | 0.02* | 34.5% |
| | *Larix* | 4.32 | 5.29 | 2.68 | 0.03* | 35.6% |
| | *Picea* | 3.8 | 4.65 | 4.20 | 0.003** | 35.7% |
| | *Pinus sibirica* | 8.62 | 8.89 | 45.23 | 0.00*** | 27.9% |
| | *Pinus sylvestris* | - | - | - | - | - |
| | Primary cover | 1.74 | 2.21 | 7.46 | 0.00*** | 33.3% |
| Shireet Naiman Nuur | *Abies* | - | - | - | - | - |
| | *Betula* | 2.57 | 3.211 | 3.82 | 0.01* | 20.7% |
| | *Larix* | 1 | 1 | 1.59 | 0.21 | 2.83% |
| | *Picea* | 1 | 1 | 6.55 | 0.01* | 9.70% |
| | *Pinus sibirica* | 3.98 | 4.91 | 4.02 | 0.003** | 27.5% |
| | *Pinus sylvestris* | 1 | 1 | 7.99 | 0.01** | 12% |
| | Primary cover | 4.01 | 4.96 | 6.38 | 0.00*** | 37.4% |
| Uggi Nuur | *Abies* | - | - | - | - | - |
| | *Betula* | 6.49 | 7.59 | 2.02 | 0.06 | 8.65% |



| | | | | | |
|---|---|---|---|---|---|
| *Larix* | 6.48 | 0.06 | 104.4 | 0.00*** | 12.2% |
| *Picea* | 1 | 1 | 0.18 | 0.67 | 0.1% |
| *Pinus sibirica* | 8.55 | 8.94 | 6.19 | 0.00*** | 19.4% |
| *Pinus sylvestris* | - | - | - | - | - |
| Primary cover | 8.07 | 8.76 | 5.72 | 0.00*** | 18.4% |

851

852