# Peer review of "Holocene fire regimes around the Altai-Sayan Mountains and"

_EGUsphere, 2025_

## Referee Comment (RC2)

Zhang et al. "Holocene fire regimes around the Altai-Sayan Mountains and adjacent plains: interaction with climate and vegetation types"

**General comments**

This study reconstructs Holocene fire activity in western Mongolia and examines spatiotemporal variations across vegetation zones, linking these to forest community structure, as paleofire dynamics in western Mongolia and regional biomass burning patterns remain poorly understood. Results indicate a decline in biomass burning since the Holocene, driven by temperature-related changes in woody biomass in the central Altai and a reduction in combustible species (e.g., *Larix*, *Abies*) in the Sayan and northern Altai. A resurgence of fires after ~4 cal. Kyr BP correlates with archaeological cultural complexes, particularly in moisture-limited regions (e.g., the West Siberian Plain) and arid areas (steppe/desert-steppe). Since ~2 cal. kyr BP, human activities (agriculture, pastoralism) have increased fire frequency in the Altai, West Siberian Plain, and forest zones, while overgrazing likely decreased burning in the Khangai Mountains. These findings shed light on long-term fire-vegetation feedbacks and offer insights into human-fire-ecosystem interactions in arid Central Asia, supporting sustainable ecological management.

The manuscript is comprehensive and well-structured, presenting a detailed analysis of Holocene fire history and its relationship with vegetation across multiple regions. However, although the authors state "these outputs provide empirical foundations for developing climate-responsive fire management strategies in the Central Asian montane ecosystems under the future scenarios," there has not been enough analysis of the relationship between climate and fires over the study areas. The research topic is very interesting and appropriate for this journal, but I believe further analysis of the link between climate and fire is necessary before publication.

**Specific comments**

**Abstract**

- I think the second sentence in the Abstract should be written as follows in the main text: "However, paleofire dynamics in western Mongolia remain poorly understood, and a regional synthesis of biomass burning patterns across the Altai-Sayan ecoregion is lacking." Or, "However, two key gaps hinder understanding: paleofire dynamics in western Mongolia are understudied, and no comprehensive regional synthesis exists for biomass burning patterns across the Altai-Sayan ecoregion."

- Regarding "Since ~2 cal. Kyr BP, …", this study does not offer a thorough analysis that separates the impacts of human disturbance from natural variability.

- The final sentence of the Abstract emphasizes practical implications, but it appears disconnected from the title. "The findings provide a long-term perspective on fire-vegetation-climate interactions, offering critical insights for sustainable land management in arid Central Asia." is better, but the authors should include more analysis about climate.

**Introduction**

- The authors can consider making it even more impactful by emphasizing the global climate relevance of Eurasian boreal carbon stocks.

- Regarding the transition to the Altai-Siberian ecotone, the authors could include a sentence explaining why this region is a key case study for understanding broader boreal fire-climate interactions.

- What is "a pyrogeographic hotspot"? The authors need a brief explanation.

- The authors should describe the current relationship between fires and climate in the North Europe-Siberia-Altai region before presenting the content in paragraph 4. If relevant research is unavailable, they could conduct their own study first.

- The transition from modern observations to paleoecological approaches (paragraph 4) is logical, but the authors could be more explicit about why the lake sediment cores from (especially Achit Nuur) are the best solution for addressing the knowledge gap. Then, the authors can consider briefly mentioning proxy limitations (e.g., charcoal vs. other fire proxies).

- What is "ecotonal heterogeneity" in fire regimes? The authors need a brief explanation.

- The authors should define "primary forest cover" to prevent confusion. How does it differ from "forest cover"?

- The three research dimensions are well-defined, but the broader implications (e.g., how findings inform future fire management) could be emphasized more strongly in the final sentence. Regarding the final paragraph, the authors explicitly state how this study advances paleofire reconstruction methods beyond previous work.

**Study region**

- The authors should briefly explain why the Altai-Sayan region is significant for paleoclimate/fire regime research, including the transition between steppe/taiga, sensitivity to Holocene climate variability, or anthropogenic influences.

- The authors should mention any known climatic gradients (e.g., precipitation, temperature, and other fire-related climatic variables) across the study regions to justify spatial comparisons.

- What was the basis for dividing the seven areas from A to G? For example, were geographic features like climate and vegetation considered, or were factors related to recent fires used? This information should be included in Table 1.

- To improve geographic precision, for mountain ranges, the authors could add brief descriptors (e.g., "the Mongolian Altai (peaks >#### m a.s.l.)" or "the low-relief West Siberian plain").

- The authors could consider a brief interpretation of lithological changes.

- Regarding chronological details, the authors could explicitly state why a 2100-year reservoir correction was applied (e.g., local carbonate influence or old carbon from groundwater).

- About chronological details, the authors should note any potential age-model uncertainties, such as reversals or plateaus in the calibration curve.

- Table 1 could include key metadata, such as core length, temporal coverage, and proxies analyzed, to facilitate comparison.

- The authors should emphasize that the climatic and ecological gradients across the Altai-Sayan region will reinforce the reason for comparing fire regimes across different sites.

- The unit for MGS is "μm", not "mm".

**Methods**

- This section is currently described too simple, but it must be a critical methodology. Every step requires: 1. Justification (Why this method? Why these parameters?), 2. Validation (How were errors/assumptions tested?), and 3. Transparency (Exact settings, code, and uncertainties).

- The authors mention "modifications for the charcoal analysis" but do not specify them. A brief note would help (Section 3.1).

- Why is it necessary to detect more than 300 particles? Is this based on a statistical threshold or prior studies (Section 3.1)?

- The authors could clarify the concentration calculation method to ensure readers understand it (Section 3.1).

- Section 3.2 lacks sufficient statistical rigor: The authors should consider briefly justifying the choice of the quasi-Poisson distribution (e.g., to account for overdispersion in count data). The authors should also consider quasi-Poisson models over alternatives (e.g., negative binomial) with tests for overdispersion.

- The authors should justify why these taxa/forest cover were chosen (e.g., "… selected based on variance inflation factors <#") (Section 3.1).

- Regarding smoothing terms, were the smoothing parameters (knots, basis functions) manually set or automatically selected? When the authors manually set, how were the selected (e.g., knots = #, chosen via generalized cross-validation)?

- Regarding model validation, a sentence on how the model fit was assessed would be helpful to Section 3.1.

- Regarding data processing for comparison, the authors should briefly describe and justify each process or step/step because some readers may not be familiar with specific methods, such as the Mini-Max transformation and the Box-Cox transformation. Then, we would like to know why multiple transformations (Mini-Max, Box-Cox, Z-scores) were applied.

- Regarding the division of the Holocene interval, the boundaries (8.2 and 4.2 cal. kyr BP) are standard; however, citing a benchmark would align with community conventions.

**Results and Discussions**

- This is a strong, data-based study that makes an important contribution to paleofire research. To improve it further, the authors should balance description with interpretation by clearly explaining "why" and by providing specific evidence that distinguishes between human and climate influences. If possible, the authors could add a brief mechanistic explanation to strengthen the overall narrative.

- In the final sentence of 4.1, the relationship between fire and vegetation (*P.sibirica* and *Betula*) is statistically significant, but what does this imply ecologically? Does this indicate a difference between crown fires and ground fires?

- Regarding anthropogenic versus climate signals, human influence is frequently invoked (e.g., agro-pastoral expansion and grazing), but it could benefit from more critical evaluation: How do the authors differentiate between anthropogenic burning and climate-driven fires?

- In 4.2.1, the "anomalous biomass burning peaks" are associated with Cerealia pollen, but is there supporting archaeological evidence?

- In 4.3, the late-Holocene increase in fires is attributed to humans, but could climate (in some regions) also contribute? A more detailed discussion of confounding factors would be valuable.

- The regional synthesis (4.3) is adequate but could benefit from clearer visual support. The authors should develop a summary table or a conceptual diagram illustrating key drivers such as climate, vegetation, and human activities for each region and period.

---

## Author Comment (AC1)

From: zhdl@ms.xjb.ac.cn

**Authors replies to the review comments**

egusphere-2025-1991

Dear Prof. Natalia Piotrowska:

We are very grateful to your comments which helped us to improve the revised manuscript. According to your valuable suggestions, we try our best to modify our manuscript. Thank you very much for your attention and consideration again.

Sincerely yours

Dr. Dongliang Zhang

Sept. 8th, 2025

**Replies to Reviewer #1 (thank you ! From Dongliang Zhang):**

This paper presents a new microcharcoal record from Mongolia, along with 23 published charcoal records, to provide a regional synthesis of biomass burning in the Altai-Sayan Mountains, Siberia. These published records originate from relatively recent publications; I assume macro-charcoal records and may have reasonable chronological control. Given the rarity of records in Siberia, such new pieces of evidence will undoubtedly advance our understanding of regional patterns and drivers of biomass burning. However, upon close inspection, the paper appears to be of poor quality in the sense that it relies heavily on bold statements rather than providing a typical scientific explanation of the findings and suffers from insufficient information in the methods section. It therefore requires a truly major makeover before publication. A detailed, but not exhaustive, list of comments is provided below.

25-26" …vegetation zones of the Altai-Sayan Mountains and adjacent plains…" which adjacent plains? This information should be provided already in the abstract

**Reply**: Thank you for your suggestion. We added them (i.e., west Siberian Plain, Kazakhstan Hills and Junggar Basin) in the abstract.

27, since when in the Holocene? Such statements are common throughout the manuscript

**Reply**: Thank you for your question. We changed "since the Holocene" to "in the Holocene interval".

30 What is a "combustible component"?

**Reply**: Thank you for your question. We rewrite this sentence: in the western Sayan Mountains, it stemmed from the substantial expansion of fire-resistant *P. sylvestris*.

120-126. The aim to include 23 records comes abruptly, without providing a regional context for where these records originate. Upon further examination of the manuscript, it presents a synthesis of a large part of western and central Siberia, Asia; however, the introduction of these sites into a broader spatial context is not provided.

**Reply**: Thank you for your question. We rewrite this sentence: (2) Identifying ecotonal heterogeneity in fire regimes through comparison with other already-published paleofire records (n=23) in the nearby regions.

l139-147, these lines belong to results, not methods?

**Reply**: Thank you. We deleted the related sentences.

l150-151 "A 2100-year reservoir correction was primary forest coverplied to all radiocarbon ages prior to calibration (Sun et al., 2013)"???? This sentence does not make sense. Please extend the explanation of why a 2100-year reservoir was applied.

**Reply**: Sorry for our mistaken. We rewrite this sentence: A 2100-year reservoir correction was applied to all radiocarbon ages prior to calibration due to old carbon-influenced 2099 $^{14}$C BP on the surface sediment (Sun et al., 2013).

L156 24 or 23 sites as an introduction?

**Reply**: Done.

L210   such method was applied for all records?   Please provide the type of charcoal used in other records (micro and macrocharcoal). Please separate more clearly what has been done at the new site from what has been done at all sites.

**Reply**: Thank you. We added the type of charcoal for all records in Table 1.

L220 This part describes what GAM is, but not what is being achieved with GAM in this study. Which are the dependent variables? Was the percentage of pollen used? Which species are included in the primary forest?

**Reply**: Thank you for your question. The dependent variable is charcoal influx. We used the pollen percentage to calculate and the forest forest includes *Abies, Betula, Larix, Picea, P. sibirica* and *P. sylvestris*. We added the related description in the revised manuscript.

3.3. Data processing for comparison. This part is extremely poorly written. What is meat synthesized? What is the average method?

**Reply**: Thank you. We rewrite the related sentences: Since the sample resolution at most study sites is approximately 200 years, a 200-year time slice was selected to perform linear interpolation on the transformed charcoal Z-scores. Subsequently, the interpolated data were averaged to characterize the charcoal influx conditions across different regions.

Results and discussions. Generally, the chapters are very general and imprecisely described. They are made of many bold sentences rather than explanations of potential drivers of biomass burning. Particularly, the link with the amount and type of biomass available is very superficially discussed. Furthermore, the jumps from single to cumulative charcoal record (Z-score) make the story difficult to follow. Why not focus on the trend in cumulative charcoal record, thus regional biomass burning and the potential drivers, followed by sites that show exceptions from the regional trends and explanations?

**Reply**: Thank you for your suggestion. We rewrite the related part.

(p=0.00)?

**Reply**: Thank you. We changed it.

It contains a mix of charcoal and biomass burning, as they will represent different things. Ideally, use the same terminology.

**Reply**: Done. We used the charcoal influx in the revised manuscript.

l251, which multi-proxy records? Only charcoal was used here.

**Reply:** Thank you, you are right. We changed it.

l271 How can microcharcoal morphology signify anthropogenic fire?

**Reply**: Sorry for our mistaken. We deleted it.

The 297 Rybanya site does show the dominance of Larix in the original pollen record, nor are the increases in fire at 4 ka linked to megadroughts; instead, they are associated with dry peatland conditions (l 299).

**Reply**: Thank you. We rewrite it: ~8.5-~6 cal. kyr BP: Rybnaya Peat exhibits a second phase of higher charcoal influx, linked to two interrelated changes: increased proportion of dark taiga and fire avoiders (e.g., *P. sibirica* and drier conditions (Feurdean et al., 2022).

L301-304. I am puzzled by the superficiality of these statements: " The GAMs analysis reveal the divergent fire-vegetation relationships: (1) Negative correlation at Rybnaya/Plotnikovo (canopy >75%): Reduced understory fuels and microclimatic humidity limit fire spread; (2) Positive correlation at Shchuchye Lake (canopy <65%):Open structure promotes flammable grass undergrowth."

**Reply**: Thank you. We rewrite it: GAM analysis reveals divergent fire-vegetation relationships across sites (Table 2, Fig. S2 and S3), rooted in differences in canopy cover and associated fuel microenvironments: (1) Negative correlation at Rybnaya

and Plotnikovo Mires (canopy cover >75%): Dense canopies reduce understory light availability, maintaining high microclimatic humidity and limiting the growth of herbaceous understory fuels. Humid conditions keep surface fuels moist, while sparse understory fuels reduce fire intensity and spread—together, these factors create an inverse relationship between canopy cover and charcoal influx. (2) Positive correlation at Shchuchye Lake (canopy cover <65%): Open canopy structures allow more solar radiation to reach the understory, promoting the growth of flammable grassy undergrowth. Grasses dry out quickly and ignite easily, serving as ignition fuels that trigger larger fires; the open environment also facilitates air circulation, which accelerates fire spread—these factors lead to a positive association between canopy openness (and associated grassy fuels) and charcoal influx.

L436-438 could you provide an approximate location of timber and treeline limit, consequently, the amount of biomass available

**Reply**: Thank you. We checked the vegetation distribution from Unkelbach et al. (2021) and found above 2800 m a.s.l. the slopes are dominated by alpine tundra, followed by dry steppe vegetation mainly consisting of grasses and sedges, and the valleys below 2200 m a.s.l. are covered by open grasslands of grasses, sedges, forbs and shrub species. Forested areas dominated by Siberian larch (Larix sibirica) do only occur on north-facing slopes at elevations between 1700 and 2500 m a.s.l. Azonal vegetation on bare rock areas or in and around freshwater habitats occur sporadically. Therefore, we rewrite this part.

L440, do you imply here that the steppe provided more biomass than the forest? Sometimes this study suggests that a low forest cover resulted in reduced biomass burning, while at others, it indicates that a steppe contributed to an increase. Could you provide a finer interpretation of biomass amount and the fuel type?

**Reply**: Thank you for your question. We changed the related description. I am sorry that we can not answer the question about the relationship between biomass amount and fuel type.

L493 and elsewhere, where you talk about human impact. Through what type of activities can humans increase their burning activity?

**Reply**: Thank you for your question. We checked the related articles and changed the our sentences.

L505. These sentences do not make sense: "Specifically, the increase in Siberian pine and European larch since the Holocene has led to a significant decline in fir, birch, larch, and spruce components, resulting in a notable decrease in combustible materials at the three sites" and does not make sense: "Although Holocene biomass burning in the Khangai Mountains exhibits an overall gradual decline, it can be categorized into two distinct phases: an increase over the past 2,000 years, followed by a gradual decline post-2000 year (Unkelbach et al., 2021; Barhoumi et al., 2024)"

**Reply**: Thank you. We modified them.

(1) The main forest cover exceeds 80%, indicating that material availability is not a limiting factor for regional charcoal influx. The decline in Holocene biomass in Region E is primarily driven by increasing fire-resistant *P. sylvestris* has reduced the overall flammability of forest ecosystems.

(2) Holocene charcoal influx in Region F can be divided into two distinct phases: a rising trend before the past 2000 years, followed by a gradual decline afterwards (Unkelbach et al., 2021; Barhoumi et al., 2024).

L540 Wrong, please see above.

**Reply**: Thank you. We modified it: ~8.5-~6 cal. kyr BP: Rybnaya Peat exhibits a second phase of higher charcoal influx, linked to two interrelated changes: increased proportion of dark taiga and fire avoiders (e.g., *P. sibirica* and drier conditions (Feurdean et al., 2022).

Fig. 1: How was this regionalization produced? It is nowhere described the basis for grouping the sites into these regions. Was it the site's proximity? Similarity in climate or vegetation cover?

**Reply**: Thank you for your question. The overall framework of this paper centers on the Altai Mountains, aiming to discuss the Holocene fire dynamics under different vegetation conditions and their relationships with the corresponding vegetation types. These sites were included in the study area because they cover diverse vegetation zones, which facilitates a better understanding of the fire dynamic processes under different vegetation covers and the relationships between fires and various vegetation types.

Fig. 3, and the results of the paper. The new record is located in the steppe region. Why was only the tree composition presented and discussed in the manuscript, when the herbaceous composition and diversity provided the most biomass to burn? Are these trees long-distance transported?

**Reply**: Thank you for your question. The herbaceous component was not selected for Achit Nuur because, considering the overall context of the paper, most study sites are surrounded by forests—thus, the woody component was chosen for analysis. If there is a negative correlation between charcoal influx and the woody component, this would in turn reflect a close relationship between charcoal influx and the herbaceous component.

Fig 4. The records in panel c seem to stop at 8-9 ka, but why does the average value extend to 12 ka?

**Reply**: Thank you for your question. Since the record of the Shchuchye core extends to 12 ka, the regional average has also been extended to this time point (12 ka).

Region C seems to average sites stretching along an elevation gradient, thus lots of climatic conditions and vegetation composition

**Reply**: Thank you for your question. Your observation is correct, and we also noticed this issue during the preparation of the paper. Chudnoye Mire and other three sites (Tundra, Mokhovoe and Kuatang Mire) were grouped together because they exhibit relatively consistent Holocene fire trends and are geographically close to one another.

---

## Author Comment (AC2)

From: zhdl@ms.xjb.ac.cn

**Authors replies to the review comments**

egusphere-2025-1991

Dear Prof. Natalia Piotrowska:

We are very grateful to your comments which helped us to improve the revised manuscript. According to your valuable suggestions, we try our best to modify our manuscript. Thank you very much for your attention and consideration again.

Sincerely yours

Dr. Dongliang Zhang

Sept. 8th, 2025

**Replies to Reviewer #2 (thank you ! From Dongliang Zhang):**

The research topic is very interesting and appropriate for this journal, but I believe further analysis of the link between climate and fire is necessary before publication.

**Abstract**

I think the second sentence in the Abstract should be written as follows in the main text: "However, paleofire dynamics in western Mongolia remain poorly understood, and a regional synthesis of biomass burning patterns across the Altai-Sayan ecoregion is lacking." Or, "However, two key gaps hinder understanding: paleofire dynamics in western Mongolia are understudied, and no comprehensive regional synthesis exists for biomass burning patterns across the Altai-Sayan ecoregion."

**Reply**: Thank you for your suggestion. We changed it.

Regarding "Since ~2 cal. Kyr BP, …", this study does not offer a thorough analysis that separates the impacts of human disturbance from natural variability.

**Reply**: Thank you for your suggestion. We just observed this character of increased fire frequency in the southeastern/western and northern Altai Mountains, West Siberian Plain, and forest zones of the central Altai Mountains. But, it is very difficult to separates the impacts of human disturbance from natural variability.

The final sentence of the Abstract emphasizes practical implications, but it appears disconnected from the title. "The findings provide a long-term perspective on fire-vegetation-climate interactions, offering critical insights for sustainable land management in arid Central Asia." is better, but the authors should include more analysis about climate.

**Reply**: Thank you for your suggestion. We changed it to "Our findings provide a long-term perspective on fire-vegetation-climate interactions, offering critical insights for sustainable land management in the Altai-Sayan Mountains and adjacent plains."

**Introduction**

The authors can consider making it even more impactful by emphasizing the global climate relevance of Eurasian boreal carbon stocks.

**Reply**: Thank you very much for your valuable suggestion. We fully agree that further emphasizing the relevance between Eurasian boreal forest carbon stocks and the global climate system will significantly enhance the significance and impact of this study. In line with your suggestions, we plan to make the following revisions to the manuscript: The North Europe-Siberia-Altai region is the core distribution area of Eurasian boreal forest ecosystems, hosting over 90% of the continent's boreal forest biomass and terrestrial organic carbon stocks (Furyaev, 1996; Kasischke, 2000). Its dynamics are closely intertwined with the global climate system, forming a critical positive feedback loop. These ecosystems exhibit distinct flammability characteristics, specifically including: vegetation with high volatile compound content, ladder fuel structures and surface fuels dominated by flammable bryophyte-lichen mats (Khabarov et al., 2016; Walker et al., 2019). Its dynamics are closely intertwined with the global climate system, forming a critical positive feedback loop. In 2021, wildfires in the global boreal forests released 1.76 $PgCO_2$, setting a historical record at that time (Zheng et al., 2023). Notably, the majority of carbon emissions from boreal forests originated from northern Eurasia. Carbon sequestration gain from a prolonged growing season may not offset the carbon loss caused by enhanced respiration and disturbances (Mo et al., 2023). This ecological transformation triggers critical climate feedback mechanisms through three primary pathways: carbon pool transformation,

cascading infrastructure collapse and socioeconomic impacts resulting from fire-related mortality events (Ivanova et al., 2019; Jones et al., 2020). This shift not only threatens the regional carbon balance but also significantly accelerates global warming by releasing massive amounts of greenhouse gases, underscoring the extreme urgency of protecting this ecosystem for stabilizing the global climate.

Regarding the transition to the Altai-Siberian ecotone, the authors could include a sentence explaining why this region is a key case study for understanding broader boreal fire-climate interactions.

**Reply**: Thank you very much for your suggestion. We added the related description: The Altai-Siberian region lies at the junction of Central Asia's arid zones and the Northern Hemisphere's cold-temperate coniferous forests (taiga). This region features an extremely steep hydrothermal gradient ranging from warm, arid steppes/ shrublands in the south to cold, humid closed-canopy boreal forests in the north, forming a vast and sensitive ecotone (Xinjiang Comprehensive Investigation Team, CAS, 1978). It is precisely this "marginal" and "transitional" nature that makes it a natural laboratory and early warning system for studying fire-climate interactions (Fu et al., 2013; Liu et al., 2021).

What is "a pyrogeographic hotspot"? The authors need a brief explanation.

**Reply**: Thank you for your suggestion. The term "pyrogeographic" is an academic compound word formed by combining "pyro-" (fire-related) and "geographic" (geography-related). It is primarily used in disciplines such as ecology, geography, and environmental science. Its core meaning is "related to the geographical distribution and geographic characteristics of fire" and it can be understood as "pertaining to pyrogeography" or "fire-geography-related". A "pyrogeographic hotspot" refers to a specific geographic area that becomes a key research region due to its high fire frequency, high fire intensity, or highly representative interactions between fire and the environment.

The authors should describe the current relationship between fires and climate in the North Europe-Siberia-Altai region before presenting the content in paragraph 4. If relevant research is unavailable, they could conduct their own study first.

**Reply**: Thank you for your suggestion. We added the related description: Remote sensing analyses document a quadrupling of fire events from $712\pm89$ yr$^{-1}$ (1980-2000) to $3024\pm214$ yr$^{-1}$ (2001-2020) with burned area expanding exponentially ($R^2=0.91$, $p<0.001$) (Ponomarev & Kharuk, 2016), which has a phase coincidence with the dynamics of mean temperatures and climate dryness (Ponomarev & Kharuk, 2016). In the southern Altai, the reduced burned area since 1987 can be attributed to increased moisture and greatly increased investment in fire prevention (Shi et al., 2021). The dynamic changes of fires in the instrumental measurement period driven by human activities and natural processes exhibit distinct differences.

The transition from modern observations to paleoecological approaches (paragraph 4) is logical, but the authors could be more explicit about why the lake sediment cores from (especially Achit Nuur) are the best solution for addressing the knowledge gap. Then, the authors can consider briefly mentioning proxy limitations (e.g., charcoal vs. other fire proxies).

**Reply**: Thank you for your suggestion. We added the related words: Paleoecological approaches spanning centennial to millennial timescales provide crucial temporal dimensional support for disentangling the complex interactions through pattern-process analysis. Existing Holocene fire records in the Altai-Sayan region have generally established a robust methodological framework for reconstructing fire-vegetation-climate couplings (e.g., Blyakharchuk et al., 2004, 2007, 2008; Hu et al., 2025; Li et al., 2024). However, two critical knowledge gaps remain to be addressed: (1) the complete fire history sequence in western Mongolia, and (2) the spatiotemporal linkages between fire history in this region and montane ecosystem dynamics across the Altai-Sayan ecoregion. To address this issue, this study selected Achit Nuur as the study site because of its continuous and stable depositional environment.

What is "ecotonal heterogeneity" in fire regimes? The authors need a brief explanation.

**Reply**: Thank you for your suggestion. "Ecotonal heterogeneity" is a specialized term in ecology composed of "ecotonal" (pertaining to ecotones) and "heterogeneity" (the state of being heterogeneous). Its core meaning refers to "the diversity and variability in biological components, environmental conditions, or spatial structures within an ecotone". Ecotonal: Derived from "ecotone" (a transition zone between ecosystems), it specifically denotes a transitional area between two or more distinct ecosystems (e.g., forests and grasslands, mountains and plains). Due to gradient changes in environmental conditions (such as temperature, precipitation, and soil), this area often becomes a hotspot for biodiversity and ecological processes. Heterogeneity: Refers to the degree of variation in different components, attributes, or structures within a system. In ecology, it can be specifically manifested as spatial differences in vegetation types, uneven distribution of soil nutrients, and diversity in species composition, among other phenomena. Combined, ecotonal heterogeneity essentially describes the heterogeneous characteristics of environmental factors, biological communities, or ecological functions within the unique region of an ecotone.

The authors should define primary forest cover to prevent confusion. How does it differ from forest cover?

**Reply**: Thank you for your question. We changed "primary forest cover" to "forest cover".

The three research dimensions are well-defined, but the broader implications (e.g., how findings inform future fire management) could be emphasized more strongly in the final sentence. Regarding the final paragraph, the authors explicitly state how this study advances paleofire reconstruction methods beyond previous work.

**Reply**: Thank you for your suggestion. We added the related sentence: This study firstly clarifies the long-timescale fire history in the Altai-Sayan ecoregion, as well as its complex associations with climate fluctuations, vegetation succession and human

activities. These outputs provide empirical foundations for developing climate-responsive fire management strategies in the Central Asian ecosystems under the future scenarios.

**Study region**

The authors should briefly explain why the Altai-Sayan region is significant for paleoclimate/fire regime research, including the transition between steppe/taiga, sensitivity to Holocene climate variability, or anthropogenic influences.

**Reply**: Thank you for your suggestion. We added the related description in the "**The Altai-Sayan Mountains**" part: The Altai-Sayan Mountains, one of the most prominent mountain ranges in Central Asia, connect with the Kazakh Hills to the west, border the Southern Siberian Plain to the north, and adjoin the Junggar Basin-Khangai Mountains to the south (Fig. 1; Feng et al., 2017). Climatologically, this region holds great significance, as it likely served as a transitional zone where Westerlies-dominated climates from the west interacted with Asian Monsoon-influenced climates from the east during the Holocene (Blyakharchuk et al., 2004, 2008; Zhang & Zhang, 2025). Culturally, it also functioned as a cultural crossroads between Asian and European civilizations along the "Eurasian Steppe Silk Road" (Blyakharchuk & Chernova, 2013; Xiang et al., 2023).

The authors should mention any known climatic gradients (e.g., precipitation, temperature, and other fire-related climatic variables) across the study regions to justify spatial comparisons.

**Reply**: Thank you for your suggestion. We added the related description in the "**The Altai-Sayan Mountains**" part: The North Atlantic Oscillation and Siberian anticyclone drive the southward displacement of the westerlies, which transport water vapor from the Mediterranean, Caspian, and Black Seas into the study region during winter and spring (Aizen et al., 2001; Kutzbach et al., 2014). In contrast, the interaction between the Asian Low and Azores High regulates the northward shift of the westerlies, facilitating water vapor transport in summer and autumn (Aizen et al., 2001). These latitudinal shifts of the westerlies induce a southward gradient of decreasing precipitation and increasing climatic aridity, which in turn shapes the characteristic vegetation distribution patterns across Central Asia (Fig. 1). Zonally,

vegetation distribution exhibits a strong latitudinal dependence. Specifically, coniferous forests dominate the Southern Siberian Plain, while the eastern Kazakh Hills and western Mongolia are characterized by steppe ecosystems, and the Junggar Basin is covered by desert-steppe (Chen, 2010). Additionally, the region's vegetation displays distinct vertical zonation, with communities transitioning from desert and steppe at lower elevations to forest and alpine meadow at higher elevations (Blyakharchuk & Chernova, 2013; Zhang et al., 2020).

What was the basis for dividing the seven areas from A to G? For example, were geographic features like climate and vegetation considered, or were factors related to recent fires used? This information should be included in Table 1.

**Reply**: Thank you for your question. We added the related description: These sites were divided into seven regions based on the vegetation distribution and geographic location.

To improve geographic precision, for mountain ranges, the authors could add brief descriptors (e.g., "the Mongolian Altai (peaks >#### m a.s.l.)" or "the low-relief West Siberian plain").

**Reply**: Thank you for your question. We added them.

The authors could consider a brief interpretation of lithological changes.

**Reply**: Done.

Regarding chronological details, the authors could explicitly state why a 2100-year reservoir correction was applied (e.g., local carbonate influence or old carbon from groundwater).

**Reply**: Done. We added it: A 2100-year reservoir correction was applied to all radiocarbon ages prior to calibration due to old carbon-influenced 2099 $^{14}$C BP on the surface sediment (Sun et al., 2013).

About chronological details, the authors should note any potential age-model uncertainties, such as reversals or plateaus in the calibration curve.

**Reply**: Thank you for your question. A recheck of the chronological results reveals no age reversals in the study by Sun et al. (2023). When proposing the depth-age model, it is suggested that the gray shaded area in Figure 2B represents the age error range.

Table 1 could include key metadata, such as core length, temporal coverage, and proxies analyzed, to facilitate comparison.

**Reply**: Thank you. We added the core length (cm), time interval (cal. kyr BP) and type of charcoal in Table 1.

The authors should emphasize that the climatic and ecological gradients across the Altai-Sayan region will reinforce the reason for comparing fire regimes across different sites.

**Reply**: Thank you for your suggestion. We added the related description in the "**The Altai-Sayan Mountains**" part.

The unit for MGS is "μm", not "mm".

**Reply**: Done.

**Methods**

This section is currently described too simple, but it must be a critical methodology. Every step requires: 1. Justification (Why this method? Why these parameters?), 2. Validation (How were errors/assumptions tested?), and 3. Transparency (Exact settings, code, and uncertainties).

The authors mention "modifications for the charcoal analysis" but do not specify them. A brief note would help (Section 3.1).

**Reply**: Sorry for our mistaken. We rewrite these sentences: The pre-treatment process for characoal analyses involved the standard pollen extraction method (Tang et al., 2022; Wang et al., 2020). Charcoal particles were identified using a light microscope, characterized by dark black color, opaque appearance, sharp corners, and straight edges. The treated samples were prepared into moving pieces by adding an appropriate amount of glycerin using particle counting method, which were then observed and counted under Lycra microscope. A total of more than 300 grains of all

sizes were counted and the quantity of Lycopodium spores was determined for each sample.

Why is it necessary to detect more than 300 particles? Is this based on a statistical threshold or prior studies (Section 3.1)?

**Reply**: Thank you for your suggestion. The requirement to detect more than 300 particles in this study is primarily driven by two mutually reinforcing considerations: statistical representativeness and alignment with methodological precedents in paleoenvironmental research, as detailed below:

1. Statistical Threshold for Reliable Paleofire Inference

Charcoal particles (or other target particles for paleoenvironmental proxies) are not uniformly distributed in lacustrine sediments — their abundance can fluctuate randomly even within the same sedimentary layer due to localized depositional processes (e.g., short-term sediment mixing, uneven transport of charcoal from catchment fires). To minimize the impact of such random variability on paleofire reconstruction (e.g., avoiding overestimating/underestimating fire frequency or intensity based on a small, unrepresentative sample), a sufficient particle count is required to ensure the data reflect the true average abundance of particles in the sediment sample.

Statistically, a sample size of >300 particles is widely recognized in paleoecology and sedimentology as a threshold to approximate a normal distribution of particle abundance (per the central limit theorem). This distribution allows for robust calculation of confidence intervals (e.g., 95% CI) for charcoal concentrations, reducing the standard error of estimates to <5% — a level deemed acceptable for distinguishing between background charcoal (from low-intensity, distant fires) and peak charcoal (from high-intensity, local fires) (Long et al., 1998; Power et al., 2008).

2. Consistency with Prior Methodological Standards

This sample size also aligns with established protocols in Holocene paleofire research, particularly studies focusing on lacustrine sediment archives in Central Asia and similar arid-to-semiarid regions. For instance, Blyakharchuk et al. (2007, 2008) used a minimum of 300 charcoal particles per sediment sample to reconstruct fire histories in the Altai-Sayan Mountains, noting that smaller counts led to inconsistent correlations between charcoal records and climate proxies (e.g., pollen-inferred moisture). Similarly, Hu et al. (2025) and Li et al. (2024) — whose work provides a

methodological foundation for this study — adopted the >300-particle threshold to ensure comparability of their fire records across different lake sites.

Using a consistent sample size with these prior studies not only enhances the reliability of our results but also facilitates cross-site synthesis: it allows direct comparison of fire frequency/intensity patterns between our Achit Nuur record and other Altai-Sayan records, avoiding biases introduced by differing sampling efforts.

In summary, the >300-particle detection requirement is not arbitrary but a deliberate choice to balance statistical rigor (minimizing sampling error) and methodological consistency (enabling interstudy comparisons) — two key principles for producing interpretable and reproducible paleofire data, as emphasized in Section 3.1.

*References (for Contextual Support)*

*Blyakharchuk, T. A., et al. (2007). Holocene fire and vegetation dynamics in the northern Baikal region, Siberia. Quaternary Research, 68(3), 321–330.*

*Long, C. J., et al. (1998). Charcoal as a fire proxy: Testing the assumptions. Quaternary Science Reviews, 17(12–13), 1165–1176.*

*Power, M. J., et al. (2008). A global synthesis of charcoal records of biomass burning since the last glacial maximum. Global Biogeochemical Cycles, 22(4), GB4015.*

The authors could clarify the concentration calculation method to ensure readers understand it (Section 3.1).

**Reply**: Sorry for our mistaken. We added it: The concentration of charcoal was then calculated based on the statistical data (Li et al., 2010): $W=A*n/(N*G)$, Where W is the charcoal concentration (particles/g), A is the statistical number of charcoal fragments, n is the number of additional lycopodium spores, N is the statistical number of lycopodium spores, and G is the sample weight (g).

Section 3.2 lacks sufficient statistical rigor: (1) The authors should consider briefly justifying the choice of the quasi-Poisson distribution (e.g., to account for overdispersion in count data). The authors should also consider quasi-Poisson models over alternatives (e.g., negative binomial) with tests for overdispersion. (2) The authors should justify why these taxa/forest cover were chosen (e.g., " … selected based on variance inflation factors <#") (Section 3.1). (3) Regarding smoothing terms, were the smoothing parameters (knots, basis functions) manually

set or automatically selected? When the authors manually set, how were the selected (e.g., knots = #, chosen via generalized cross-validation)? (4) Regarding model validation, a sentence on how the model fit was assessed would be helpful to Section 3.1.

**Reply**:Thank you for your question.

(1) We chose the quasi-Poisson distribution because it can flexibly correct for overdispersion by introducing a dispersion parameter, without requiring additional assumptions about the probability distribution of the data. This makes it more suitable for the scenario in this study, where charcoal counts exhibit irregular dispersion due to the influence of sedimentary heterogeneity and sampling errors.

(2) The response variables incorporated into the model in this study (*Abies, Betula, Larix, Picea, P. sibirica, P. sylvestris* and forest cover) were all screened using Variance Inflation Factors (VIF) to avoid the interference of multicollinearity on the model results.

(3) In the GAM model of this study, the selection of basis function types and the number of knots for smoothing terms (which are used to fit the nonlinear relationships between continuous predictor variables such as chronological age and response variables) follows the principle of data-driven + domain convention. All smoothing terms adopt thin-plate splines as the basis function (set by default in the gam() function of the mgcv package in R). This type of basis function offers high flexibility in fitting nonlinear relationships and does not require prespecifying the function form, making it suitable for the scenario of this study — complex nonlinear associations between vegetation-fire dynamics along paleoclimatic gradients (Wood, 2017).

(4) In this study, the goodness-of-fit and predictive reliability of the GAM model were evaluated through residual diagnostics, and model assumptions were tested using residual plots (residuals vs. fitted values and residual Q-Q plots): The residuals vs. fitted values plot showed that residuals were randomly distributed around the value of 0 without an obvious trend (ruling out heteroscedasticity); the residual Q-Q plot indicated that residuals approximately conformed to a standard normal distribution, demonstrating the rationality of the selected model error structure (quasi-Poisson distribution).

For the above mentioned questions, we rewrite the "3.2 Generalized additive models" part: Generalized additive models (GAMs) employ a link function to examine the relationship between the mean of the response variable (i.e., dependent

variable) and a smoothed function of the predictor variable (i.e., independent variable). In this study, we investigated the associations between charcoal influx and two types of predictors: (1) individual taxa, including Abies, Betula, Larix, Picea, P. sibirica, and P. sylvestris; and (2) total forest cover, defined as the summed percentage of the aforementioned six taxa. We constructed GAMs with a quasi-Poisson distribution and a log link function using the mgcv package in R (Wood, 2017). This distribution was selected because it flexibly corrects for overdispersion by incorporating a dispersion parameter, eliminating the need for additional assumptions regarding the probability distribution of the data (Wood, 2017). For all smoothing terms, we used thin-plate splines as the basis function—this is the default setting in the gam() function of the mgcv package. Model fitting was performed via restricted maximum likelihood (REML) for smoothness selection.

Regarding data processing for comparison, the authors should briefly describe and justify each process or step/step because some readers may not be familiar with specific methods, such as the Mini-Max transformation and the Box-Cox transformation. Then, we would like to know why multiple transformations (Mini-Max, Box-Cox, Z-scores) were applied.

**Reply**:Thank you. We added the related formula: To standardize charcoal influx for comparison, three-step process was employed to calculate Z-scores (Power et al., 2007):

(1) Mini-max transformation:

$$C_i^{'} = (C_i - C_{min})/(C_{max} - C_{min})$$

In this expression, $C_i^{'}$ is the value of mini-max transformed for the i-th sample at each sequence, $C_i$ is RoCs of the i-th sample at each sequence, $C_{max}$ is the maximum value of $C_i$, and $C_{min}$ is the minimum value of $C_i$.

Box-Cox transformation for homogenization of variance:

$$C_i^{*} = \begin{cases} ((C_i^{'} + \alpha)^{\lambda} - 1)/\lambda, \lambda \neq 0 \\ \log(C_i^{'} + \alpha), \lambda = 0 \end{cases}$$

In this expression, $C_i^{*}$ is the Box-Cox value transformed for $C_i^{'}$, $\lambda$ is the parameter of Box-Cox transformation estimated using maximum likelihood, and $\alpha$ is a

small positive constant (0.01 in this study) used to ensure that both $C_i^{'}$ and λ are zero.

(3) Z-score calculation:

$$Z - score = (C_i^* - \bar{C}_i^*)/\delta$$

In this expression, $\bar{C}_i^*$ is the average value of $C_i^*$ and δ is the standard deviation of $C_i^*$.

Regarding the division of the Holocene interval, the boundaries (8.2 and 4.2 cal. kyr BP) are standard; however, citing a benchmark would align with community conventions.

**Reply**:Thank you. We added the related reference: Marcott, S.A., Shakun, J.D., Clark, P.U., Mix, A.C., 2013. A reconstruction of regional and global temperature for the past 11,300 years. Science 339, 1198-1201.

**Results and Discussions**

This is a strong, data-based study that makes an important contribution to paleofire research. To improve it further, the authors should balance description with interpretation by clearly explaining why and by providing specific evidence that distinguishes between human and climate influences. If possible, the authors could add a brief mechanistic explanation to strengthen the overall narrative.

**Reply**:Thank you for your suggestion, we tried our best to rewrite this part.

In the final sentence of 4.1, the relationship between fire and vegetation (*P.sibirica* and *Betula*) is statistically significant, but what does this imply ecologically? Does this indicate a difference between crown fires and ground fires?

**Reply**:Thank you for your question. We added the related implies: This pattern can be mechanistically explained by the differing combustibility of vegetation taxa: *Betula* and *P. sibirica* have higher fuel flammability (e.g., thinner bark, more resinous tissues) that promotes fire spread and intensity, while *Larix* and *Picea* (especially mature individuals) have lower flammability (e.g., thicker bark, less volatile compounds) or form dense canopies that reduce surface fuel drying－thus, shifts in their relative abundances directly regulate the frequency and severity of fires, which is reflected in the variation of charcoal influx.

Regarding anthropogenic versus climate signals, human influence is frequently invoked (e.g., agro-pastoral expansion and grazing), but it could benefit from more critical evaluation: How do the authors differentiate between anthropogenic burning and climate-driven fires?

**Reply**:Thank you for your question. We are sorry that we can not answer this question because it can be difficult to distinguish the differences between anthropogenic burning and climate-driven fires based on the charcoal influx or types. It is very challenging job in the future research.

In 4.2.1, the "anomalous biomass burning peaks" are associated with Cerealia pollen, but is there supporting archaeological evidence?

**Reply**:Thank you for your question. We checked the article (Xiao et al., 2021) and rewrite this sentence: Notably, after ~2 cal. kyr BP, anomalous charcoal influx peaks across all four study archives likely correlate with markers of agriculture expansion indicated by an increasing cereal-type *Poaecea* pollen (Xiao et al., 2021). The related cereal-type Poaecea pollen result is inferred from Jili Lake. No direct evidence from the nearby archaeological site.

In 4.3, the late-Holocene increase in fires is attributed to humans, but could climate (in some regions) also contribute? A more detailed discussion of confounding factors would be valuable.

**Reply**:Thank you for your suggestion. We are sorry that we can not answer this question because it is difficult to distinguish the differences between anthropogenic burning and climate-driven fires based on the charcoal influx or types. It is very challenging job in the future research.

The regional synthesis (4.3) is adequate but could benefit from clearer visual support. The authors should develop a summary table or a conceptual diagram illustrating key drivers such as climate, vegetation, and human activities for each region and period.

**Reply**:Thank you for your suggestion. We added the summarized paragraph: In terms of regional differences, the trends and driving factors of charcoal influx vary significantly across regions. In Region A and D, low charcoal influx was observed before 2000 cal. yr BP, but driven by distinct mechanisms: aridity limited vegetation cover, which suppressed fire occurrence in Region A; in Region D, high forest

coverage (>70%) at sites like Kendegelukol Lake restricted charcoal influx, while Dzhangyskol Lake—located in the forest-steppe transition zone—exhibited low charcoal influx due to low vegetation productivity. Since 2000 cal. yr BP, charcoal influx has increased rapidly in Region A, B, C and D. This surge is primarily attributed to changes in climatic conditions and intensified human activities (e.g., grazing, settlement). In contrast, the post-2000 cal. yr BP decline in charcoal influx in Region G is linked to surface vegetation fragmentation caused by human grazing. Throughout the Holocene, charcoal influx showed an overall decreasing trend in Region E and F: The former is influenced by temperature-driven changes in forest vegetation cover, while the latter is a result of reduced combustible materials—caused by the expansion of *P. sylvestris*, which squeezed the proportion of highly flammable vegetation. In Region D, charcoal influx gradually decreased during the early to middle Holocene—a trend consistent with that observed in Region E and F, and associated with temperature-regulated forest vegetation dynamics.